# Sequential addition of neuronal stem cell temporal cohorts generates a feed-forward circuit in the *Drosophila* larval nerve cord

Yi-wen Wang[1†], Chris C Wreden[1†], Maayan Levy[2], Julia L Meng[3], Zarion D Marshall[4], Jason MacLean[2,4,5,6], Ellie Heckscher[1,2,3,5,6]*

[1]Department of Molecular Genetics and Cell Biology, University of Chicago, Chicago, United States; [2]Committee on Computational Neuroscience, University of Chicago, Chicago, United States; [3]Program in Cell and Molecular Biology, University of Chicago, Chicago, United States; [4]Committee on Neurobiology, University of Chicago, Chicago, United States; [5]Department of Neurobiology, University of Chicago, Chicago, United States; [6]University of Chicago Neuroscience Institute, Chicago, United States

**\*For correspondence:**
heckscher@uchicago.edu

[†]These authors contributed equally to this work

**Competing interest:** The authors declare that no competing interests exist.

**Abstract** How circuits self-assemble starting from neuronal stem cells is a fundamental question in developmental neurobiology. Here, we addressed how neurons from different stem cell lineages wire with each other to form a specific circuit motif. In *Drosophila* larvae, we combined developmental genetics (twin-spot mosaic analysis with a repressible cell marker, multi-color flip out, permanent labeling) with circuit analysis (calcium imaging, connectomics, network science). For many lineages, neuronal progeny are organized into subunits called temporal cohorts. Temporal cohorts are subsets of neurons born within a tight time window that have shared circuit-level function. We find sharp transitions in patterns of input connectivity at temporal cohort boundaries. In addition, we identify a feed-forward circuit that encodes the onset of vibration stimuli. This feed-forward circuit is assembled by preferential connectivity between temporal cohorts from different lineages. Connectivity does not follow the often-cited early-to-early, late-to-late model. Instead, the circuit is formed by sequential addition of temporal cohorts from different lineages, with circuit output neurons born before circuit input neurons. Further, we generate new tools for the fly community. Our data raise the possibility that sequential addition of neurons (with outputs oldest and inputs youngest) could be one fundamental strategy for assembling feed-forward circuits.

## Editor's evaluation

The article presents a thorough analysis of specific neuronal lineages in the early larval ventral nervous system that relates the birth order to circuit connectivity and function. The key findings of the work are (1) the identification of sharp temporal cohort divisions for the lineages under investigation, (2) synapse formation between neurons of different lineages and temporal cohorts, and (3) the observation that output neurons in this instance are born prior to input neurons.

## Introduction

Neuronal circuits are the fundamental functional units of the nervous system, and neuronal stem cell lineages are the fundamental developmental units. Determining lineage–circuit relationships is

**eLife digest** The nervous system of an animal consists of complex arrangements of nerve cells or neurons. These arrangements are called neuronal circuits, and they contain both input and output neurons. Input neurons sense signals, such as external cues, and output neurons pass these signals on to the brain, for example. The nerve cells in a circuit connect to each other through so-called synapses in specific patterns. Neuronal circuits first assemble during the development of an animal. The assembly process starts when a nerve stem cell divides and gives rise to more specialized neurons, its progeny.

A lot of what we know about neuronal circuit assembly comes from studying the nerve cord of fruit fly larva, which shares many features with the spinal cord of vertebrates. Previous studies had used experimental techniques to trace, or follow, the fate of the progeny of specific nerve stem cells. These approaches provided information about which nerve stem cells contribute to which neuronal circuits. However, major questions in developmental neurobiology remain about how exactly these neuronal circuits assemble. For example, it was not clear in what order input and output neurons build a circuit.

Here Wang, Wreden et al. took a different approach by starting with a specific circuit in the fruit fly nerve cord – a circuit that detects vibrations – and looking for the stem cells contributing to that circuit. Using a number of techniques, Wang, Wreden et al. determined when particular nerve cells were 'born', what they looked like, and what other nerve cells they formed synapses with.

Although nerve stem cells gave rise to many different neurons during development, those neurons did not change gradually over time. Instead, neurons were born in short bursts, and those in the same 'temporal cohort' were similar to each other, while neurons in different cohorts were different. The neuronal circuit that detects vibrations assembled itself from three temporal cohorts of neurons coming from different stem cells. The output neurons, which send information from the nerve cord to the brain, were born before the input neurons, which detect vibrations in the surroundings.

All in all, these experiments offer more detailed insights into how neuronal circuits assemble during development. The study also provides experimental resources for other scientists working with fruit flies, and poses new research questions for developmental biologists studying vertebrates.

essential for deciphering the developmental logic of circuit assembly (*Li et al., 2018*; *Meng and Heckscher, 2021*). So far, lineage–circuit relationships have been described in an ad hoc, lineage-by-lineage basis in a handful of model circuits. For example, in the vertebrate neocortex, excitatory neurons from a single stem cell preferentially populate individual neocortical microcolumns (*Yu et al., 2009a*). In the vertebrate hippocampus, excitatory neurons from a single stem cell share input (*Xu et al., 2014*). In the insect visual system, synchronous production of neurons from a single stem cell underlies the retinotopy of direction-selectivity (*Pinto-Teixeira et al., 2018*). Together, these studies demonstrate that lineage–circuit relationships differ depending on circuit anatomy. However, it remains unknown how neurons from different stem cell lineages wire with each other to form specific circuit motifs. In this study, we took a new approach, the converse of previous studies, starting from circuit and identifying the developmental origins of the neurons. This allows us to identify developmental rules governing circuit self-assembly.

As a model, we use the *Drosophila* larval nerve cord. Circuits in the nerve cord, like those in the spinal cord, processes multiple somatosensory stimuli and generate patterned muscle contractions (*Meng and Heckscher, 2021*). In addition, spinal cords and nerve cords share gross morphology (left-right symmetrical, segmented), and both contain neurons with shared function and homologous gene expression (*Heckscher et al., 2015*; *Catela and Kratsios, 2021*). However, only for the nerve cord is a connectome available. This connectome is a high-resolution, transmission electron microscopic image of all central nervous system (CNS) neurons and synapses (*Ohyama et al., 2015*). The larval connectome enables anatomical circuit tracing at cellular and synaptic resolution. Nerve cord stem cells (neuroblasts) in *Drosophila* larvae are extremely well-characterized (*Doe, 2017*). In each segment, there are 30 left–right pairs of neuroblasts (*Figure 1A*), and combinatorial expression of spatial transcription factors (e.g., row and column genes) make each neuroblast unique (*Broadus et al., 1995*). Further, each neuroblast produces a unique, invariant lineage of neurons (e.g., *Figure 1B*; *Schmid et al., 1999*). In general, neuroblasts divide multiple times using a type 1 division pattern. In type 1

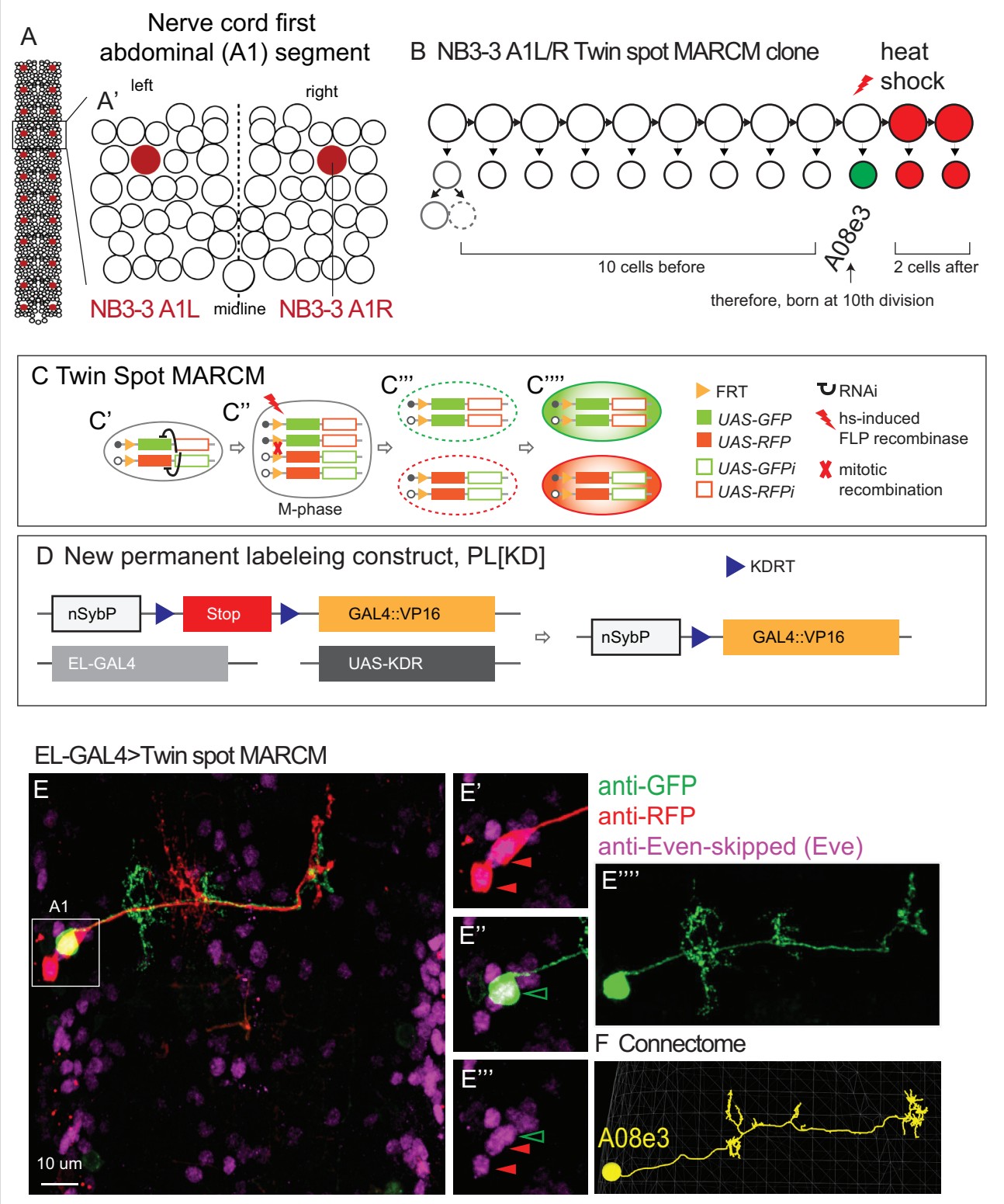

**Figure 1.** Twin-spot mosaic analysis with a repressible cell marker (ts-MARCM) determines the birth order and morphology of NB3-3A1L/R neurons. (**A, B**) Illustrations of *Drosophila* neuroblasts. (**A**) The nerve cord is left–right symmetrical and segmented. Each circle represents one neuroblast with NB3-3 in maroon. Segment A1 (boxed) is enlarged in (**A'**). It contains 30 types of neuroblasts. (**B**) NB3-3 lineage progression is shown with an example ts-MARCM clone overlaid. Each circle represents one cell and each arrow represents a cell division. First, NB3-3 divides to self-renew and generate a ganglion mother cell, which divides to generate a motor neuron (solid circle) and an undifferentiated cell (dashed circle). Then, NB3-3 directly generates

*Figure 1 continued on next page*

*Figure 1 continued*

ELs. In ts-MARCM, a heat shock is provided (red lightning bolt) as NB3-3 divides. In this example, a singly labeled neuron is shown in green (A08e3), and two alternatively labeled neurons are shown in red. Because the total number of neurons in the lineage is known, counting labeled neurons allows inference of neuronal birth order. The identity of the singly labeled neuron is determined by matching the labeled neuron to the corresponding neuron in the connectome using morphological criteria (see 'Materials and methods'). (**C, D**) Illustration of ts-MARCM genetic constructs used in this study. Our updated version of ts-MARCM system has four components (**C, D**). (1) It uses a pair of genetically modified chromosomes. On one chromosome is an FRT recombinase site (yellow triangle) followed by a *UAS-GFP* (solid green box) and a *UAS-RFP-RNAi* (hollow red box) construct. On the other chromosome is an FRT site followed by a *UAS-RFP* (solid red box) and a *UAS-GFP-RNAi* (hollow green box) construct. When cells are heterozygous for these chromosomes, the GFP- and RFP- RNAi constructs ensure repression of GFP and RFP protein expression, respectively (black curves, **C'**). (2) It has a heat-shock-inducible FLP recombinase (red lightning bolt). By varying the heat shock protocol, we control both the timing and amount of FLP supplied. Heat shocks induce FRT-based chromosomal recombination in dividing cells (red X, M-phase cell, **C''**). A subset of recombination events produce a pair of post-mitotic progeny, one of which is homozygous for the *UAS-GFP, UAS-RFP-RNAi* construct, and the other homozygous for the *UAS-RFP, UAS-GFP-RNAi* construct. In these cells, RNAi is no longer able to repress GFP or RFP expression (**C'''**). (3) A cell-type-specific GAL4 line, (e.g., *EL-GAL4,* light gray box in **B**) is used to drive expression of *UAS-RFP* or *UAS-GFP* (**C''''**). (4) To get robust ts-MARCM labeling in early-stage larvae, it was often necessary to amplify GAL4 expression. To do so, we generated a new permanent labeling construct (**D**). Specifically, a neuron-specific nSyb promoter (white box) is upstream of a Stop (red box) flanked by KDRT (blue triangles) recombination sights. When the KDR recombinase (from UAS-KD, dark gay box) is supplied, the Stop is removed, and nSyb drives expression of a the new GAL4 (yellow box). This new GAL4 is the GAL4 DNA binding domain tethered to the strong transcriptional activator VP16. (**E, F**) Image of a ts-MARCM clone and a corresponding neuron in the connectome. (**E**) Many segments of the nerve cord are shown in dorsal view with anterior up. The boxed region in segment A1 is enlarged at the right. In this ts-MARCM clone, two neurons are labeled in red and one in green (arrowheads), and all are Eve(+) ELs. The singly labeled EL is enlarged to highlight morphological detail. The corresponding neuron in the connectome is shown in (**F**). Specific genotype is listed in ***Supplementary file 4***.

The online version of this article includes the following figure supplement(s) for figure 1:

**Figure supplement 1.** NB3-3-GAL4 line.

divisions, the neuroblast divides asymmetrically to self-renew and generate a ganglion mother cell (GMC). The GMC divides to produce two neurons (***Figure 1B***), which are differentiated from each other by Notch signaling (***Skeath and Doe, 1998***). All Notch ON neurons populate an A hemilineage, and all Notch OFF neurons populate a B hemilineage (***Truman et al., 2010***). Some neuroblasts switch to a type 0 division pattern (e.g., ***Figure 1B***). In type 0 divisions, the neuroblast divides asymmetrical to self-renew and to generate a Notch OFF neuron (B hemilineage) (***Baumgardt et al., 2014***). Within a hemilineage, neurons have unique temporal identities, or birth orders (e.g., first-born, second-born, etc.). Temporal identity is specified by temporal transcription factors (***Isshiki et al., 2001***). In summary, neuronal stem cells, the lineages of neurons they produce, and the molecular mechanisms underlying the generation of neuronal diversity are extremely well characterized in the *Drosophila* nerve cord.

It has been suggested that the genetic mechanisms underlying specification of neuronal diversity also govern circuit formation and function (***Mark et al., 2021***; ***Sagner and Briscoe, 2019***). Evidence to supports this idea comes from the identification of temporal cohorts in the *Drosophila* larval nerve cord. Temporal cohorts are sets of neurons in a hemilineage that are born within a tight time window. Initially, temporal cohorts were correlated with circuit membership using functional approaches. Specifically, neuroblast NB3-3 generates a series of EL interneurons (*E*ven-skipped [eve] neurons with *L*aterally placed cell bodies) (***Wreden et al., 2017***). Molecularly, ELs are subdivided into early-born EL and late-born EL temporal cohorts based on the expression of an enhancer called 11F02 (***Wreden et al., 2017***). EL interneurons of the early-born temporal cohort respond to vibrational stimuli, whereas EL interneurons of the late-born temporal cohort do not (***Wreden et al., 2017***). Further, activation of early-born ELs triggers escape rolling, whereas activation of late-born ELs alters left–right symmetrical crawling (***Wreden et al., 2017***). Together, these data linked temporal cohorts with differential circuit function. More recently, in additional hemilineages (i.e., NB1-2, NB2-1, NB3-1, NB4-1, NB5-2, NB7-1, NB 7-4), neurons within temporal cohorts were shown to have similar synaptic partnerships (***Meng et al., 2019***; ***Meng et al., 2020***; ***Mark et al., 2021***). Moreover, the number of neurons in a hemilineage that are segregated into a given temporal cohort could be altered by manipulating temporal transcription factor expression in neuroblasts (***Meng et al., 2019***; ***Meng et al., 2020***). Thus, temporal cohorts are developmental units likely specified early during neurogenesis by molecular programs known to generate neural diversity.

Notably, however, previous studies that described temporal cohorts lacked the resolution to distinguish between graded and sharp wiring transition models. For many neuronal features, such as morphology, gene expression, and neurotransmitter phenotype, there can be sharp changes in

lineage progression. For example, a neuroblast can abruptly change from producing motor neurons to interneurons (*Meng et al., 2020*) or from producing Eve(+) to Eve(−) neurons (*Pearson and Doe, 2003*). The idea that temporal cohorts have distinct circuit-level function suggests that there are also sharp changes in the patterns of synaptic partnerships formed by neurons in a lineage, and that these sharp changes are correlated with temporal cohort borders. However, available data are also consistent with an alternative, graded transition model. In the graded transition model, during lineage progression, changes in wiring would slowly transition such that neurons with more similar birth times would have more similar synaptic partnerships. Distinguishing between graded and sharp transition models is fundamental for understanding lineage progression in *Drosophila* neuroblasts.

The overall objective of this article is to test the hypothesis that *Drosophila* larval nerve cord circuits are assembled by preferential connectivity between distinct temporal cohorts. Although an attractive hypothesis, there is limited supporting evidence (*Meng et al., 2019*; *Mark et al., 2021*; *Meng et al., 2020*). Before we can address this hypothesis, in the first part of this article, we needed to distinguish between graded and sharp wiring transition models. Using lineage tracing, connectomics, and network science-based statistical analysis, we find support for a sharp transition model. And so, the first part of this article brings into alignment the concepts of temporal cohorts, circuit function, and circuit anatomy at single-neuron resolution. In the second part of this article, we use the circuit containing the early-born EL temporal cohort as a model. Connectomics and calcium imaging show that early-born ELs are embedded in a feed-forward circuit (e.g., A connects to B and to C, B also connects to C). This circuit is found in the nerve cord, and it encodes the onset of vibration stimuli. The early-born ELs are the outputs of the circuit, transmitting information to the brain via their long projection axons. Next, we combine lineage tracing, single-neuron labeling approaches, and connectomics to identify the developmental origins for the majority of major interneuron inputs (i.e., those synapsing 10 or more times) onto early-born ELs. Specifically, we identify the neuroblast parent, birth order within a lineage, and birth time relative to early-born ELs. Our data support the hypothesis that this feed-forward circuit is assembled by preferential connectivity between distinct temporal cohorts from different lineages. We find connectivity among temporal cohorts from the same lineage in different anterior–posterior segments. For neurons from different lineages, our data show that circuit outputs (early-born ELs) are born before circuit inputs (i.e., Ladders, Basins). Ultimately, these data show that sequential addition of temporal cohorts, with outputs oldest and inputs youngest, is one strategy for assembling feed-forward circuits.

## Results

### NB3-3 lineage contains two temporal cohorts, early-born ELs and late-born ELs, which are distinctive both in morphology and connectivity

This study aims to understand how neurons from different stem cell lineages wire with each other to form a specific circuit motif. In particular, we wanted to test the hypothesis that *Drosophila* larval nerve cord circuits are assembled by preferential connectivity between temporal cohorts (*Meng and Heckscher, 2021*). First, however, we needed to better understand the relationship between temporal cohorts, circuit function, and circuit anatomy. Specifically, we needed to distinguish between graded and sharp wiring transition models (see 'Introduction). To do so, we need to determine the relationship between neuronal birth order and synaptic connectivity at single neuron resolution.

As a model, we use the NB3-3 lineage in the first abdominal segment (A1) of the nerve cord (*Figure 1A*). In A1, there is a left–right pair of NB3-3 neuroblasts that are thought to be identical. We will refer to these neuroblasts as NB3-3A1L/R. First, NB3-3A1L/R undergoes a type 1 division to self-renew and generate a GMC (*Baumgardt et al., 2014*). The GMC divides to produce two neurons of unknown function (*Figure 1B*; *Schmid et al., 1999*). Next, NB3-3 begins to divide using a type 0 division pattern to self-renew and generate EL interneurons (*Figure 1B*; *Baumgardt et al., 2014*). The birth timing of ELs has been characterized using three different methods (*Tsuji et al., 2008*; *Wreden et al., 2017*; *Mark et al., 2021*). Nonetheless, the precise relationship between morphology and birth order for NB3-3A1L/R progeny remains unknown. Here, we determined the birth order and morphology of every neuron produced by NB3-3A1L/R. Using morphology, we identified matching neurons in the connectome. Using the connectome, we found all upstream synaptic partners. Finally, we analyzed the patterns of wiring using network science approaches.

## Birth order and morphology of each neuron in the NB3-3A1L/R lineage

First, we needed to determine birth order and morphology of EL interneurons at the single-neuron level. To do so, we used the gold standard in the field for lineage tracing, Twin-spot mosaic analysis with a repressible cell marker (ts-MARCM) (*Yu et al., 2009b*). Briefly, in ts-MARCM, heat shocks are delivered to dividing cells, rendering their progeny competent to express either a UAS-red or -green fluorescent reporter (*Figure 1B and C*). Reporter expression is driven by a cell-type-specific GAL4 line. Providing heat shocks at various times is used to reconstruct lineage progression. The morphology of singly labeled neurons is used to determine neuron identity (*Figure 1E''''*) and counting the number of alternatively labeled, subsequently born cells determines birth order (*Figure 1E'–E'''*). To drive ts-MARCM reporter expression, we used *EL-GAL4* that drives in all EL neurons, together with a newly generated genetic cassette that amplifies GAL4 expression (*Figure 1D*). We find that in segment A1, ELs are produced in the following order: A08x, A08j1, A08j3, A08j2, A08m, A08o, A08c, A08s, A08e3, A08e1, and A08e2 (*Figure 2*). Notably, all these ELs have been suggested to be NB3-3A1L/R progeny, and our data provides the first confirmation for several (A08o, A08j1-j3) (*Mark et al., 2021*; *Wreden et al., 2017*; *Heckscher et al., 2015*). Because we already knew the first two neurons in the lineage were non-ELs, our data precisely define the birth order for all neurons in NB3-3A1L/R lineage at cellular resolution (*Figure 2*).

Next, we needed to find all NB3-3A1L/R neurons in the larval connectome because this provides high-resolution pictures of neuron morphology, including the locations of dendrites (i.e., regions with postsynaptic densities, *Figure 3D*, cyan) and axons (i.e., regions with presynaptic active zones, *Figure 3D*, red). To locate neurons of interest in the connectome we used two criteria: first, lineage-related neurons often have cell bodies that cluster and neurites that form a tight bundle entering the neuropil (i.e., the axon, dendrite and synapse-rich, cell body-free region of the CNS) (*Schmid et al., 1999*; *Mark et al., 2021*). We generated a new *NB3-3-GAL4* line, which selectively labels NB3-3 (*Figure 1—figure supplement 1*). Using both *NB3-3-GAL4* and *EL-GAL4* to drive membrane GFP, we confirmed that the GFP-labeled neurons have clustered soma and bundled neurites (*Figure 3A and B*). Second, we used ts-MARCM data as ground truth to identify neurons with corresponding morphologies in the connectome (*Figures 1–2*). With these two criteria, we identified a lineage bundle that contained all the ELs plus two other non-EL neurons (*Figure 3C*). The non-EL neurons were an undifferentiated neuron, which has never been reported before, and a motor neuron, which is consistent with previous reports (i.e., MN 22/23, *Figure 3D*, top row; *Schmid et al., 1999*). Then, with the high-resolution morphology data, we asked to what extent is EL morphology correlated with birth order and temporal cohort borders. We found one morphological group of early-born ELs, which included A08x, A08j1, A08j3, A08j2, and A08m (*Figure 3D*, second row). The axons of these ELs project to the central brain, with one exception, A08m, which projects to the thorax. Their dendrites are found on both sides of the midline (ipsilateral and contralateral to the soma) with one exception, A08m, which has only contralateral dendrites. All project dendrites ventrally from the main neurite, and of these, two had been previously identified as early-born ELs (A08x and A08m) (*Wreden et al., 2017*). The second morphological group contained late-born ELs and included A08o, A08c, A08s, A08e3, A08e1, and A08e2 (*Figure 3D*, third and fourth rows). The axons of these ELs project to the central brain or remain local. Their dendrites are ipsilateral to the soma and project dorsally from the main neurite, with one exception, A08c, which projects dendrites both dorsally and ventrally. All of these ELs, except A08o, had been previously identified as late-born ELs (*Wreden et al., 2017*). Thus, we find morphological groupings of neurons in the NB3-3A1L/R lineage that are strongly correlated with neuron birth time and that reflect previous grouping of ELs into early-born and late-born temporal cohorts.

In a previous report, Mark and colleagues used the length of a neurite between the soma and the point at which it enters the neuropil as a proxy for neuronal birth order (*Mark et al., 2021*). This measure is called 'cortex neurite length.' Here, we sought to quantify the relationship between cortex neurite length and neuron birth order because for the first time we had a complete lineage with both precisely defined birth order information and cortex neurite length data. We plotted cortex neurite length for each neuron in the NB3-3A1L/R lineage as well as their average and found a strong positive correlation as measured by Pearson's correlation (r(11) = 0.88, p<0.0001 and r(11) = 0.85, p=0.002 and r(11) = 0.92, p<0.0001 for A1L, A1R, and average, respectively; *Figure 3E*, *Figure 3—source data 1*). However, exact birth order cannot be accurately inferred by neurite length. We conclude that it is

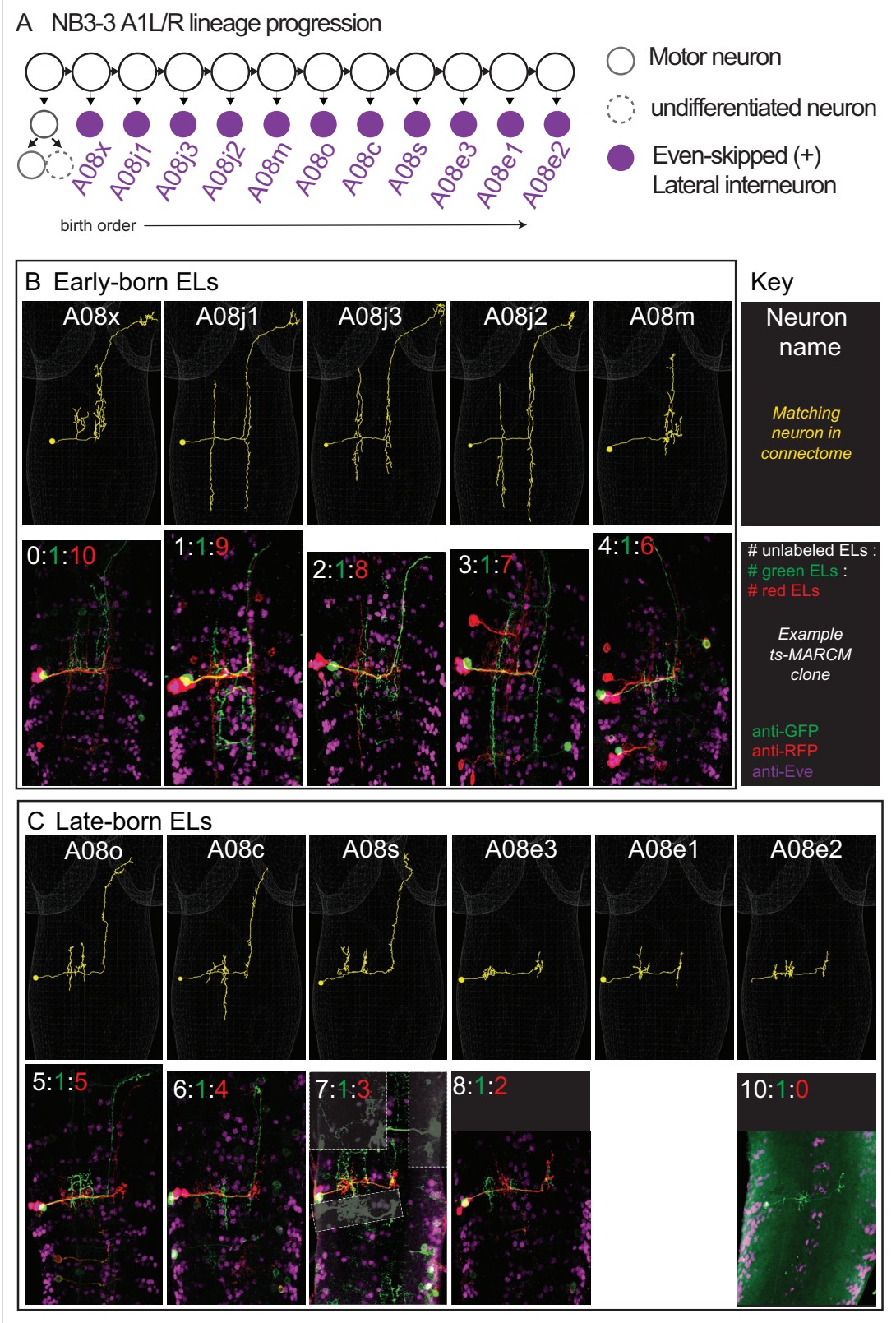

**Figure 2.** Twin-spot mosaic analysis with a repressible cell marker (ts-MARCM) provides birth order for all neurons in the NB3-3A1L/R lineage. (**A**) Schematic of NB3-3A1 lineage progression is shown with EL birth order. Each circle represents one cell, and each arrow represents a cell division. (**B, C**) Images of individually labeled ts-MARCM clones shown in birth order. Early-born ELs are show in the top box, and late-born ELs are shown in the bottom. An image key is shown to the right of the early-born ELs. Briefly, all images are shown in a dorsal view with anterior to the top. The neuron

*Figure 2 continued on next page*

*Figure 2 continued*

name is at the top of each image pair, along with an example of the neuron in the connectome (yellow). The bottom of each image pair is an example of a clone stained with anti-GFP, anti-RFP, and anti-Eve. At the top of the example clone image is the number of unlabeled ELs (white), the number of ELs labeled in green, and the number of ELs labeled in red. Sometimes clones in other segments are lightly boxed over for clarity. For genotype, see *Supplementary file 4*.

---

possible to roughly, but not precisely, infer birth order of neurons within a lineage using only anatomical data.

## Synaptic inputs onto NB3-3A1L/R neurons

Next, to characterize the relationship between neuronal birth order and patterns of synaptic connectivity, we needed to comprehensively identify all the synaptic inputs to NB3-3A1L/R neurons. To do so, we mined the connectome and identified 4944 synaptic inputs on NB3-3A1L/R neurons, which came from 1179 different neurons (or neuronal fragments) (*Figure 4—source data 1*). We categorized the sources of NB3-3A1L/R synaptic inputs: a majority (61%) came from other nerve cord interneurons, followed by sensory neurons (19%), with the remainder being from difficult to annotate neuron fragments or neurons of the central brain (*Figure 4A*, *Figure 4—source data 1*). Because a large majority of inputs onto NB3-3A1L/R neurons come from sensory neurons and nerve cord interneurons, we focused on these in the rest of this article.

A major functional difference between early-born ELs and late-born ELs is their response to sensory stimuli (i.e., early-born ELs respond to vibration, but late-born ELs do not) (*Wreden et al., 2017*). First, we characterized the relationship between birth order and sensory neuron input. In the NB3-3A1L/R lineage, the earliest born, non-EL neurons get no input from sensory neurons (*Figure 4B*, *Figure 4—source data 2*). All early-born ELs receive input from chordotonal sensory neurons (CHOs), which agrees with previous findings (*Wreden et al., 2017*) and characterizes the sensory input onto A08j1-3 for the first time (*Figure 4B*). All late-born ELs get input from proprioceptive and other sensory neurons, but not CHOs (*Figure 4B*). This agrees with previous finding (*Heckscher et al., 2015*) and newly shows that A08o gets a low amount of proprioceptive input (*Figure 4B*). Thus, we conclude that based on input from sensory neurons, there are two sharp transitions in NB3-3A1L/R lineage progression, from non-EL neurons to early-born ELs and from early-born ELs to late-born ELs.

Next, we asked if there were similarly sharp transitions in NB3-3A1L/R lineage progression when looking at inputs from nerve cord interneurons. However, first, we wanted to understand how many synaptic contacts each interneuron made onto neurons in the NB3-3A1L/R. This is because neurons contributing few synapses may represent biological noise (promiscuous wiring) or technical noise (annotation errors). In our dataset, the number of contacts from one interneuron onto a NB3-3A1L/R neuron varied from 1 to 28 synapses. To examine the full distribution of interneuron inputs onto neurons inNB3-3A1L/R, we generated a histogram with number of neurons on the Y-axis and number of synapses onto a NB3-3A1L/R on the X-axis (*Figure 4C*, *Figure 4—source data 3*). A majority (79%) of interneurons contributed only one synapse onto an NB3-3A1L/R neuron (*Figure 4C*). To avoid analyzing potentially noisy data, we focused on the subset of nerve cord interneurons for which left–right (L/R) pairs formed a minimum of two synapses onto a NB3-3A1 neuron on one side and four synapses onto the corresponding NB3-3A1 on the other side (see 'Materials and methods' for details, *Figure 4—source data 4*). We identified 198 'L/R paired' interneurons, which together provided 1990 synapses onto NB3-3A1L/R neurons. From these L/R paired interneurons, undifferentiated neuron gets no synaptic input, and MN22/23 gets inputs that largely do not synapse onto ELs (*Figure 4D*). For early-born ELs, many L/R paired interneurons synapse onto multiple neurons within the temporal cohort, but not other neurons in the lineage (*Figure 4D*). Similarly, for late-born ELs, L/R paired interneurons synapse onto multiple neurons within the temporal cohort, but not other neurons in the lineage (*Figure 4D*). We conclude that based on L/R paired interneuron input, NB3-3A1L/R lineage progression undergoes sharp transitions in patterns of input connectivity. Further, these transitions are correlated with temporal cohort borders.

The qualitative analysis above suggested that early-born and late-born ELs have distinct connectivity patterns. To quantitatively test this suggestion, we analyzed all input data, not selected subsets, using a network science-based approach, specifically distance analysis. Briefly, we treated inputs to a neuron as a vector that contained the number of synaptic contacts. The order of inputs was the

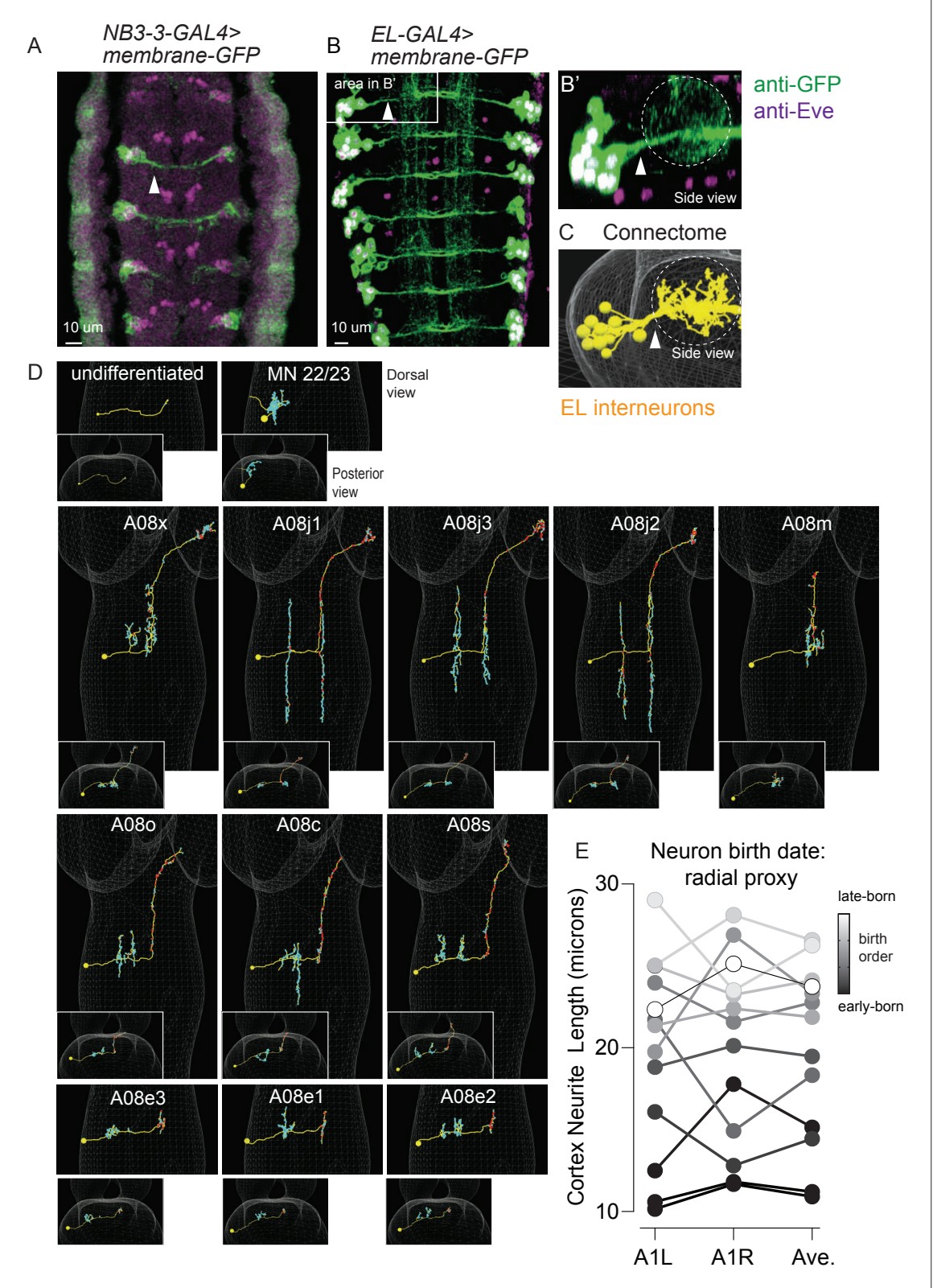

**Figure 3.** The morphology of all NB3-3A1L/R neurons in the connectome. (**A–C**) Images of NB3-3A1L neurons with clustered soma and bundled neuritis. (**A, B**) *NB3-3-GAL4* or *EL-GAL4* driving expression of membrane GFP was immortalized using a permanent labeling strategy (***Supplementary file 4***). Arrowheads point to bundles formed by neurons before they enter the neuropil. The images are dorsal views with anterior up. In (**A**), a stage 12 embryo, all NB3-3 neuronal progeny (including two non-ELs) are in a bundle. In (**B**), a first instar larva shows that in larval stages ELs form a bundle. The

*Figure 3 continued on next page*

*Figure 3 continued*

box shows segment A1L, which is enlarged and rotated in (**B′**). (**B′**) shows a posterior view of the bundle as it enters the neuropil (dashes). An image corresponding to (**B′**) from the connectome is shown in (**C**). (**D**) Images of each NB3-3A1L neuron in the connectome with synapse locations. For each image, a faint white mesh shows the outline of central nervous system (CNS) volume. Large images are dorsal views with anterior up. Smaller images are posterior views (looking towards the brain) with dorsal up. Yellow circles and lines are soma and neurites, respectively. Red and cyan dots are pre- and postsynaptic specializations, respectively. Neuron names are shown at the top of each panel. First-born NB3-3A1L/R progeny are in the top row; early-born ELs in the middle row; and late-born ELs in the bottom two rows. (**E**) Quantification of NB3-3A1L/R cortex neurite length as a proxy for birth timing. The length of the neurites between the soma and neuropil has been used as a proxy for neuronal birth timing. Plotted on the y-axis are cortex neurite lengths computed for NB3-3 neurons in segment A1L, A1R, and their average. Each dot represents a single neuron (or average of two). Gray scale shows the precise birth order as determined by twin-spot mosaic analysis with a repressible cell marker (ts-MARCM). There is a rough correlation between birth order and neurite length, with earlier-born neurons possessing shorter neurites.

The online version of this article includes the following source data for figure 3:

**Source data 1.** NB3-3A1L/R cortex neurite length.

same for every neuron. For every pair of neurons, we calculated the Euclidean distance between the two vectors, which produced a metric for the difference in connectivity between two neurons. We call this the 'real' value. For the distance of the real value to be zero, not only would the input partners need to completely overlap, but the number of synapses contributed by each of these partners would need to be identical. To provide an estimate of how the same set of neurons could be wired by chance, we computationally permutated the input vector for each neuron independently, preserving number of inputs, but shuffling input identity. We did this 100 times for each neuron to produce a 'shuffled' dataset. For shuffled datasets, Euclidian distances were computed. Real values were normalized to shuffled values to generate a z-score that allowed us to determine statistical significance (see 'Materials and methods'). In examining the distribution of z-scores, early-born ELs have significantly *similar* patterns compared to other neurons in their temporal cohort, but not compared to other neurons in the lineage (*Figure 5A*, *Figure 5—source data 1*). Late-born ELs have significantly *similar* patterns compared to other neurons in their cohort, but not compared to other neurons in the lineage (*Figure 5A*). Importantly, in general, early-born and late-born ELs have significantly *different* connectivity from each other (*Figure 5C*, *Figure 5—source data 2*). This provides statistical support for the idea that early-born ELs and late-born ELs have sharp transitions in patterns of connectivity beyond what is expected by chance. Further, these sharp transitions in connectivity correlate with previously characterized circuit function and temporal cohort borders.

It has been suggested that within a lineage temporal cohorts could be copies of each other, implying that although the specific input partners differ, the pattern of connectivity repeats (*Wreden et al., 2017*). Distance analysis allowed us to asses this idea by helping to visualize connectivity patterns between early-born versus late-born ELs (*Figure 5A*). To quantify patterns of connectivity we measured two features—similarity among left–right pairs of neurons and similarity among following pairs. An example of similarity among left–right pairs is A08j3 A1L and A08j3 A1R, both of which have significantly similar connectivity compared to A08x A1L (*Figure 5A*). In contrast, A08c A1L, but not A08c A1R, has significantly similar connectivity compared to A08o A1L (*Figure 5A*). In general, early-born ELs have significant similarities between left–right pairs of neurons (*Figure 5B*, open purple circles), whereas late-born ELs did not ( *Figure 5B*, open magenta circles). An example of similarity among following pairs is that A08c A1L has significantly similar connectivity compared to A08s A1L, which is adjacent to A08c in birth order (*Figure 5A*). Both early-born and late-born ELs have similarities among following pairs (*Figure 5B*, filled circles). The idea that neurons within a temporal cohort are more similar to neurons next born in the sequence shows that *within* a temporal cohort there are graded transitions in connectivity patterns. Additionally, early-born ELs have similarities with many neurons within their temporal cohort, whereas late-born neurons only tend to be similar to neurons that were born next in the sequence (*Figure 5A*). We conclude that temporal cohorts are not merely copies of each other and the computations performed by early-born and late-born ELs may differ.

Finally, we used distance analysis to ask what drives the clustering of EL interneurons into two cohorts—connectivity with sensory neurons, interneurons, or both? We repeated the distance analysis, dropping out either sensory neuron or interneuron inputs. For drop out of interneurons, only sensory neuron connections remain (*Figure 5D*, *Figure 5—source data 3*). In this case, there were still significant similarities in inputs specifically between neurons of a temporal cohort. However, for early-born

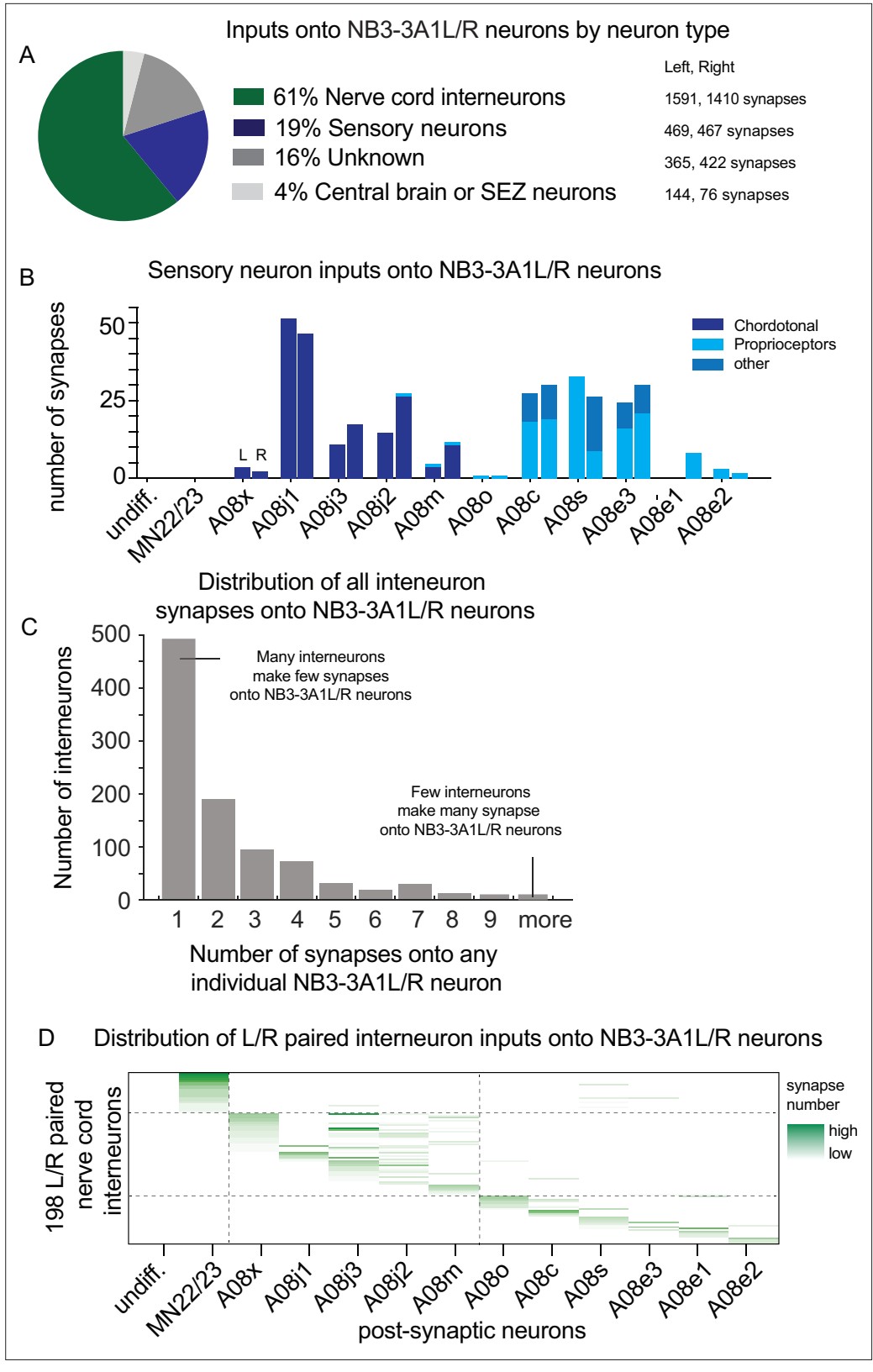

**Figure 4.** There are sharp transitions in lineage progression with respect to sensory neuron and interneuron inputs onto NB3-3A1L/R. (**A–D**) Quantification of synaptic inputs onto NB3-3A1L/R. (**A**) The pie chart displays the percentage of total synapses from a given neuron type onto NB3-3A1L neurons. Data are color-coded, and the total number of synapses contributed by a given neuron type is shown at right. (**B**) Sensory neurons form synapses

*Figure 4 continued on next page*

*Figure 4 continued*

onto NB3-3A1L/R neurons. Chordotonal sensory neurons (dark blue) synapse onto early-born ELs. Late-born ELs get input from proprioceptive or other sensory neurons (lighter blues). The Y-axis represents the proportion of total synaptic input onto a given neuron that come from a single class of sensory neuron. Columns in the X-axis represent pairs of postsynaptic NB3-3A1L (L) and NB3-3A1R (R) neurons sorted by birth order. (**C**) Many individual nerve cord interneurons synapse onto NB3-3A1L/R neurons only a few times, whereas few individual nerve cord interneurons provide multiple synapses onto NB3-3A1L/R neurons. The X-axis shows histogram bins representing number of synapses contributed to an individual neuron. The Y-axis represents the number of individual nerve cord interneurons in each bin. (**D**) Distribution of major interneuron inputs onto NB3-3A1L/R neurons. The Y-axis represents 198 left right (L/R) paired nerve cord interneurons (see 'Materials and methods'). Each column represents one neuron. Names are not shown due to space limitations. Columns in the X-axis represent postsynaptic NB3-3A1L/R neurons. If an input interneuron forms a synapse with a NB3-3A1L/R neuron, the row–column intersection is shaded green, with darker the green representing greater number of synapses. Dashed lines are placed at the border between non-ELs, early-born ELs, and late-born ELs.

The online version of this article includes the following source data for figure 4:

**Source data 1.** All synaptic input neuron information onto NB3-3A1L/R.

**Source data 2.** Sensory input onto NB3-3A1L/R neurons.

**Source data 3.** Summary of NB3-3A1L/R input neurons based on number of input.

**Source data 4.** NB3-3A1L/R input from L/R paired neurons.

ELs, the total number of neuron pairs with significant similarities was increased, whereas for late-born ELs, it was reduced. This suggests that interneuron input diversifies the early-born cohort, but unifies the late-born cohort. Data for drop out of sensory neurons is consistent with this idea (*Figure 5E*, *Figure 5—source data 4*). This may hint at differing developmental strategies for temporal cohort assembly.

In summary, we find that the NB3-3A1L/R lineage undergoes two sharp transitions during lineage progression—from non-EL neurons to early-born ELs and from early-born ELs to late-born ELs. Sharp transitions are seen with respect to morphology and patterns of input connectivity, supporting a sharp transitions model. Further, sharp transitions are correlated with previously defined early-born and late-born ELs temporal cohort borders (*Wreden et al., 2017*). Ultimately, these data bring into alignment the concepts of temporal cohorts, circuit function, and circuit anatomy at single-neuron resolution.

## Early-born ELs are embedded in a feed-forward circuit and encode the onset of vibrational stimulation

For the rest of this study, we focus on early-born ELs as a model to understand circuit assembly. Early-born ELs and late-born ELs contribute to different circuits (*Wreden et al., 2017*). In comparison to late-born ELs, the anatomical circuit motif in which early-born ELs are embedded is poorly characterized (*Heckscher et al., 2015*; *Mark et al., 2021*). Stem cell lineage-to-circuit relationships can differ depending on the type of circuit generated (*Xu et al., 2014*). And so, we needed to characterize the early-born EL circuit.

To understand the circuit to which early-born ELs contribute, we started by grouping neurons that synapse onto early-born ELs in A1L/R into classes based on morphology. We analyzed 1945 synapses formed on early-born ELs by 331 neurons (sensory neurons and nerve cord interneurons) (*Supplementary file 2*, *Figure 6—source data 1*). We grouped neurons into classes based on morphology. We calculated the total number of neurons within a class and the percentage of total synaptic input onto early-born ELs provided by neurons in each class. 6% of input came from 21 Basin interneurons (*Figure 6A and E*). Basins are a previously described class of excitatory neurons that have ipsilateral axons and dendrites (*Ohyama et al., 2015*). 12% of input are from 20 A08 interneurons (*Figure 6B and E*). In experiments described below, we find that A08 interneurons are early-born ELs from segments other than A1. 17% of synaptic inputs come from 32 Ladder interneurons (*Figure 6C and E*). Ladders are a previously described group of unpaired inhibitory neurons with cell bodies in the midline and left–right symmetrical arbors (*Jovanic et al., 2016*). 31% of synaptic inputs come from 72 CHOs (*Figure 6D and E*), which respond to vibration (*Wreden et al., 2017*; *Ohyama et al., 2013*). The remaining 34% of inputs come from 186 other interneurons, which fall into a wide variety of

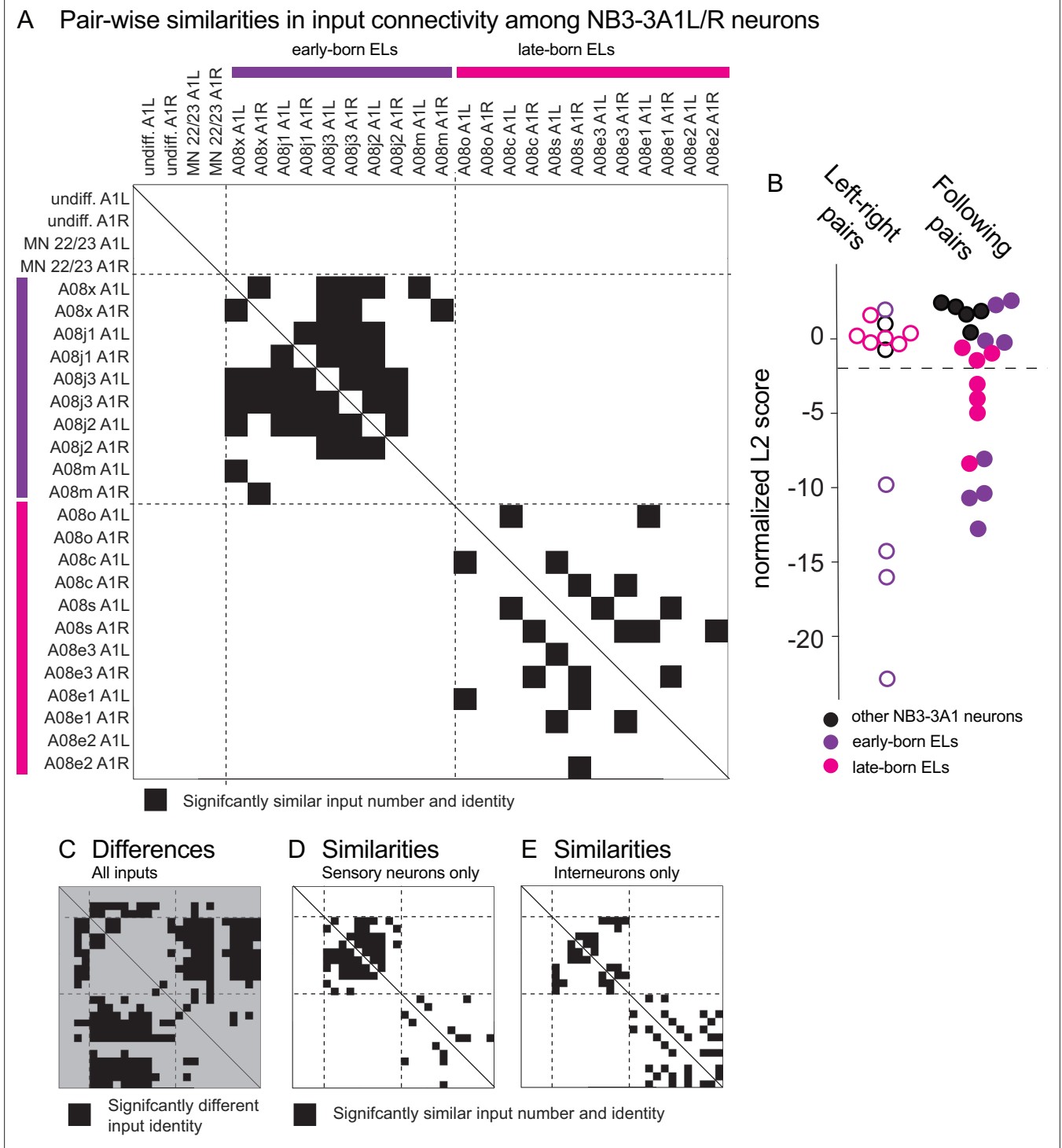

**Figure 5.** Early-born and late-born ELs have significantly similar input connectivity patterns between different temporal cohorts. (**A–E**) Quantifications of input connectivity between NB3-3A1L/R neurons using network analysis. (**A**) The plot shows pairwise comparisons of input connectivity with significant similarities in black (normalized, nonbinary distance analysis, see 'Materials and methods'). The plot is symmetric, and a solid line shows the diagonal. Left and right NB3-3A1 neurons are arranged in order of their birth on the X-axis and Y-axis. Early-born ELs are indicated with purple and late-born ELs with magenta. Row–column pairs with scores significantly smaller than shuffled controls are shown in black (p<0.05). Dashed lines are placed at the border between non-ELs, early-born ELs, and late-born ELs. (**B**) The plot shows a summary of pairwise differences in input connectivity (normalized, binary distance analysis, see 'Materials and methods'). At left is a comparison of left–right neuron pairs of the same neuron (hollow circles) and at right is a comparison of two neurons that follow each other birth order (solid circles). All left–right pairs with significant similarities are early-born ELs. Both early-born and late-born ELs have significant similarities between following pairs. Significance threshold (p<0.05) is marked in a dashed line. Smaller scores

*Figure 5 continued on next page*

*Figure 5 continued*

indicate increased input similarity. (**C**) The plot is similar to that in (**A**), but shows normalized Euclidean distance *differences* between neuron pairs. The background is gray to visually distinguish it from similarity plots. Black area indicates pairs whose input identities are significantly different than would be expected by chance (p<0.05). (**D, E**) The plots are similar to that in (**A**), but computed separately with sensory neuron-only or interneuron-only inputs.

The online version of this article includes the following source data for figure 5:

**Source data 1.** z-scores for similarity.

**Source data 2.** z-scores for differences.

**Source data 3.** Similarity z-scores for sensory only input.

**Source data 4.** Similarity z-scores for interneuron only input.

morphological classes (*Figure 6E*). In summary, in these four morphological classes, 145 neurons provide a majority of the total synaptic input (66%) onto early-born ELs in segments A1L/R.

Not only do neurons in the four classes (CHO, Ladder, Basin, EL) provide a majority of total synaptic input onto early-born ELs, but on a per neuron basis, these neurons tend to make more synapses onto early-born ELs. This is important from a functional perspective because neurons that provide multiple synaptic contacts onto a downstream are thought to be more effective in driving downstream neuronal activity. To investigate this quantitatively, we grouped CHOs, Ladders, Basins, and ELs into one group and all other neurons into a second group. We generated histograms for each group, binning by number of synapses onto early-born ELs, and counting the total number of neurons in each bin (*Figure 6F*, *Figure 6—source data 2*). Comparing these two histograms shows a significant difference in the distribution of synapse number (Kolmogorov–Smirnov, p<0.0001). For example, 36% of all neurons in the group containing CHOs, Ladders, Basins, and ELs, provide 10 or more synapses onto early-born ELs. For other neurons, it is 5%. Put another way, of all the neurons that provide 10 or more synapses onto early -born ELs in A1, >80% are CHOs, Ladders, Basins, or early-born ELs. We conclude that, in general, individual neurons in this group are more highly connected with early-born ELs in A1 compared to other neurons, and neurons in this group are also more likely to be important early-born EL activity.

After identifying these four classes of inputs, we next asked how they were anatomically arranged at the level of circuit motif. We consider early-born ELs to be the circuit outputs because they are excitatory and project to the central brain (or, for A08m, thorax) (*Figure 7A–D*, *Figure 7—source data 1*). Input to the circuit comes from CHOs, which provide *direct* excitatory input onto early-born ELs (yellow arrows, *Figure 7A–D*). CHOs also provide *indirect* input excitatory and inhibitory onto early-born EL in A1 via Ladders, Basins, and other early-born ELs (*Figure 7A–D*, orange, blue, and purple arrows, respectively). Thus, anatomically, early-born ELs in A1 are imbedded in a feed-forward circuit motif.

Because early-born ELs get direct excitatory and indirect inhibitory input from CHOs, this suggests that early-born ELs could be activated upon initial chordotonal stimulation and inactivated with delay. In previous calcium imaging experiments, we used a slow, presynaptic calcium sensor and found that early-born ELs respond to vibration (*Wreden et al., 2017*). Here, to get a more precise understanding of dynamics of EL activity, we monitored stimulus encoding using a cytoplasmic calcium sensor with faster signal decay. In response to vibration, which activates CHOs (*Wreden et al., 2017*), EL activity initially peaks and then the signal rapidly declines, although the vibrational stimulus remains (*Figure 7E–G*). These data support the idea that functionally at least a subset of ELs encode stimulus onset.

In summary, we find that all early-born ELs are embedded in a feed-forward circuit motif and that at least a subset of ELs encode the onset of vibrational stimuli.

## Selective wiring between temporal cohorts

After characterizing the circuit containing early-born ELs, we were in the position to test the hypothesis that this feed-forward circuit is assembled by preferential connectivity between distinct temporal cohorts (*Meng and Heckscher, 2021*). To test this, we needed to know to what extent could Basin, Ladder, and A08 interneurons be considered temporal cohorts. To be a temporal cohort, interneurons must be lineage-related and born within a tight time window. We used a combination of light microscopy and connectomics to address this question.

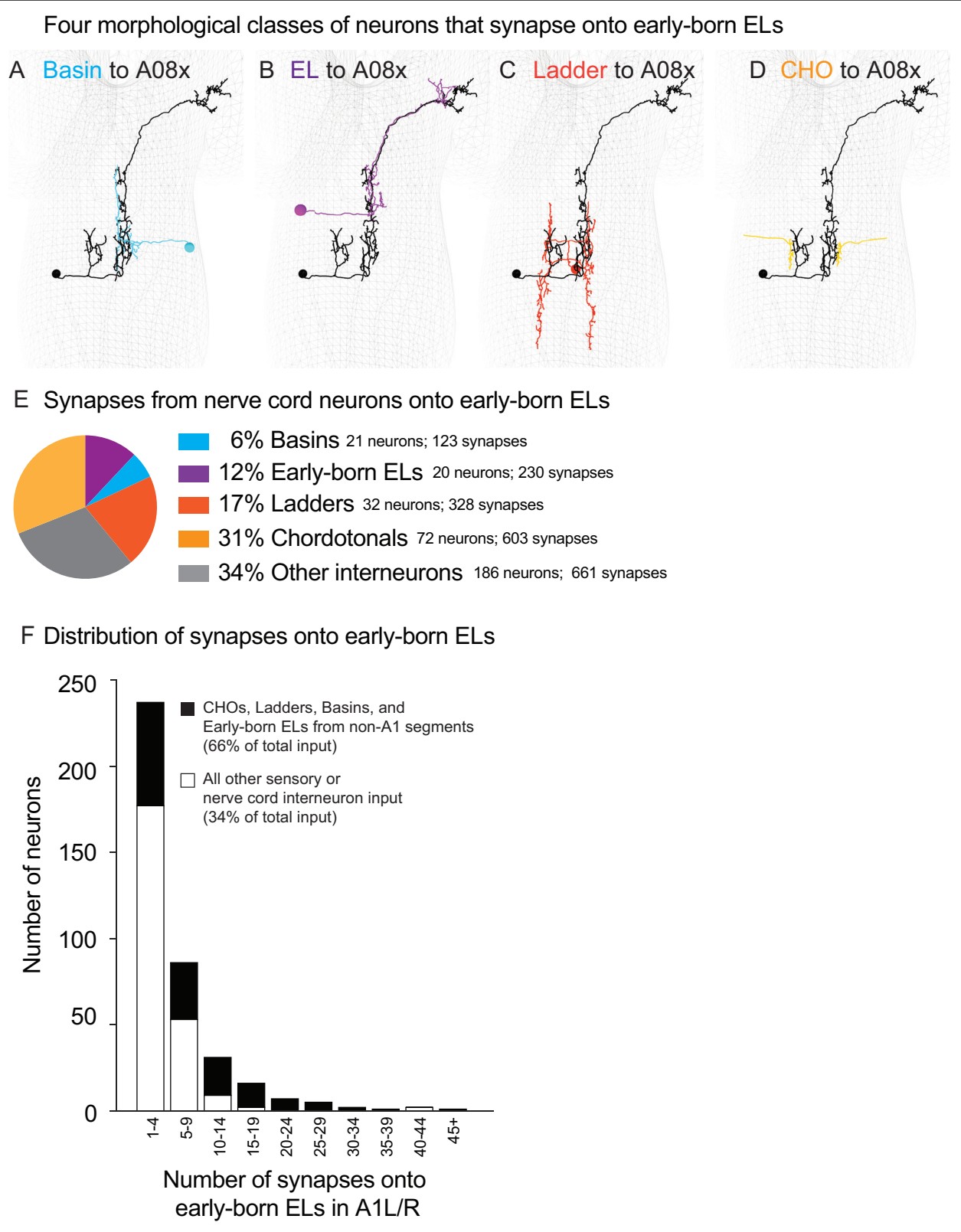

**Figure 6.** At the population level and single-neuron level, four morphological classes of neurons provide the majority of synaptic input onto early-born ELs in A1. (**A–D**) Images of neurons from each class. Examples of neuron morphology from connectome data. Black mesh shows the outline of the central nervous system (CNS), anterior is up, images show in dorsal view. The early-born EL, A08x A1L, is shown in black. The other neurons with inputs onto A08x are shown in color code in (**E**). CHO, chordotonal sensory neuron. (**E, F**) Quantification of nerve cord synapses onto early-born ELs.

*Figure 6 continued on next page*

*Figure 6 continued*

(**E**) The pie chart displays inputs onto early-born ELs as a proportion of total population of inputs. Each slice is the percentage of synapses from a given anatomical class (e.g., Ladders) compared to the nerve cord neuron synapses (*Supplementary file 2*). (**F**) A stacked histogram shows the distribution of synaptic contacts made by individual neurons. Black shows the group of neurons that includes in CHO, Ladder, Basin, and early-born ELs from segments other than A1 (black). White shows a group containing all other neurons. The X-axis are histogram bins representing a range of synapse numbers contributed to all early-born ELs in segment A1. The Y-axis represents the number of individual nerve cord interneurons in each bin.

The online version of this article includes the following source data for figure 6:

**Source data 1.** Basin, EL, Ladder, and CHO input onto early-born ELs in A1L/R.

**Source data 2.** Distribution of input onto early-born ELs in A1L/R.

## Basins are a middle-to-late-born temporal cohort from NB3-5

There are four pairs of Basin interneurons per segment, Basins 1–4 (*Ohyama et al., 2015*; *Jovanic et al., 2016*). Basins have been suggested to be a temporal cohort from NB3-5 (*Wreden et al., 2017*). NB3-5 is one of the first neuroblasts to form in *Drosophila* embryos (*Broadus et al., 1995*). It is one of the largest embryonic lineages, reportedly producing up to 36 neurons, a subset of which are found at larval stages (*Figure 8A*; *Monedero Cobeta et al., 2017*; *Moris-Sanz et al., 2014*). The first-born neurons express the CCAP neuropeptide, and many NB3-5 neurons die during embryogenesis (*Monedero Cobeta et al., 2017*; *Moris-Sanz et al., 2014*). Otherwise, the cell types produced by NB3-5 are poorly understood.

First, we asked if Basins are likely to be a lineage-related set. To do so, we used ts-MARCM and detected ts-MARCM recombination events using *Basin-GAL4*, which is expressed in all Basin interneurons (*Ohyama et al., 2015*). First, we induced ts-MARCM clones using brief heat shocks at early times in development. This produced one or two clones per CNS (e.g., *Figure 8B*) consistent with the idea that recombination levels were low. We counted the number of Basin neurons in each clone and found that most often all four Basins were labeled (*Figure 8B' and K*). This is consistent with the idea that recombination occurred in a single dividing neuroblast and that neuroblast produces all Basin interneurons. Next, we used ts-MARCM to determine the relative birth timing of Basins within the lineage. In this set of experiments, we provided heat shocks at a variety of developmental time points and scored the resulting progeny. We find that Basins are born in the following order: Basins 2, 1, 3, 4 (*Figure 8A–J*). Additionally, based on the types of clones produced at different times (see 'Materials and methods' for details), our data suggest Basins are born within a middle-to-late window during NB3-5 lineage progression (*Figure 8K*). We conclude that Basins are a lineage-related set of neurons born within a tight time window, consistent with the idea that Basins are a temporal cohort.

Next, we wanted to directly test the idea that Basins are progeny of NB3-5. To do so, we crossed a fly line expressing membrane-GFP in all Basins (*Figure 9A*) to three different fly lines, each expressing nuclear localized (nls) RFP in the NB3-5 lineage neurons. In all cases, we observed co-expression of Basin and NB3-5 lineage markers (*Figure 9B–D'*). We conclude that Basins are progeny of NB3-5.

Basin neurons had already been identified in the connectome (*Ohyama et al., 2015*). And so, next, we used connectome analysis as an independent means of characterizing the development of the Basins. First, we observe Basin cell bodies are clustered and their neurites form a bundle both in light microscopy images and in the connectome (*Figure 9A and E*). This provides further support for the idea that Basins are a lineage-related set of neurons. Second, we assayed Basin birth order using cortex neurite length as a proxy. To do so, in connectome, we identified all neurons in the lineage bundle that contained Basin neurons (*Figure 9E*, yellow) because these neurons are likely to be part of the NB3-5 lineage. Then, we measured and plotted the cortex neurite lengths of putative NB3-5 neurons in the hemilineage that included Basins. In this set, Basins possess cortex neurite lengths that are among the longest (*Figure 9H*). These data are consistent with our ts-MARCM data, which suggest that Basins are middle-to-late born. Therefore, by two independent means we find that Basins are a lineage-related set of neurons born within a tight time window. This provides strong evidence that Basins are a temporal cohort.

In the first section of this article, we found that between neurons in the NB3-3A1L/R lineage there are sharp transitions in input connectivity patterns correlated with temporal cohort boundaries. However, it remained unclear if this was true for early-born and late-born temporal cohorts only, or if this represented a more general 'rule.' Because we identified Basins as a temporal cohort, we

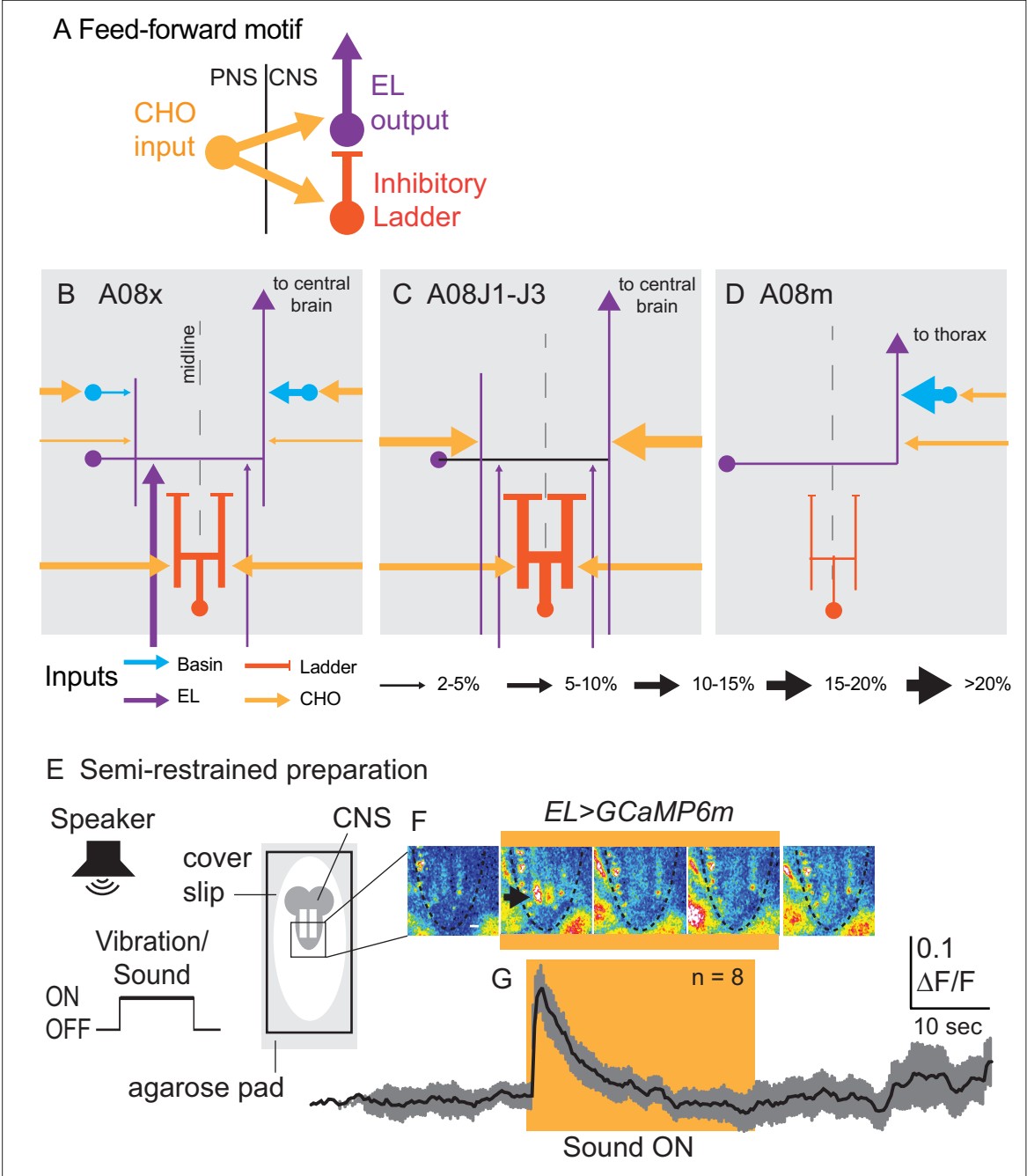

**Figure 7.** Early-born ELs are embedded in a feed-forward motif and encode the onset of vibrational stimuli. (A–D) Illustrations of patterns of synaptic inputs onto early-born ELs. (A) A simplified schematic show the feed-forward motif in which early-born ELs are embedded. (B–D) Schematics of patterns of synaptic inputs onto different early-born ELs. Arrows are excitatory connections and bars are inhibitory (i.e., Ladders). The key at bottom shows how line thickness corresponds to percentage of inputs onto a given neuron. (E) Illustration of the preparation and stimulus protocol used for calcium imaging. A larva is placed on a bed of agarose with a cover slip on top. Fluorescence in the central nervous system (CNS) (gray lobed structure with two white lines [neuropil]) is recorded before, during, and after a sound is played from a speaker. (F) Images from representative recordings of calcium signals in ELs. Frames from a representative recording are shown at 8 s intervals. Yellow box indicates frames where sound stimulus was ON. Images are pseudo-colored with white/red as high fluorescence intensity and blue as low. Anterior is up. Scale bar is 50 μm. Dashed lines show the outline of the nerve cord. The black arrow in the second image panel indicates a region of neuropil with increased fluorescence. (G) Quantifications of EL calcium signals. Changes in EL calcium signaling upon vibrational stimulus (yellow box) show a rapid increase followed by decay. The black line represents average fluorescence intensity and gray represents standard deviation. ΔF/F is the change in fluorescence over baseline. N = number of larvae recorded. For genotype, see *Supplementary file 4*.

The online version of this article includes the following source data for figure 7:

*Figure 7 continued on next page*

*Figure 7 continued*

**Source data 1.** Breakdown of Basin, EL, Ladder, and CHO input onto individual early-born ELs.

next examine the relationship between patterns of connectivity and Basin temporal cohort borders. As described in the first section of this article, using distance analysis, we quantified the significant similarities in neuronal input, this time analyzing neurons in the Basin hemilineage (*Supplementary file 3*). We find that Basins have statistically significant similarities in input connectivity with themselves compared to most other NB3-5 neurons (*Figure 9G and H*, *Figure 9—source data 1–3*). Thus, there are sharp transitions in connectivity patterns correlated with Basin temporal cohort borders (see 'Discussion' for more).

## Ladders are a late-born temporal cohort from MNB

Ladders have been suggested to be progeny of midline neuroblast (MNB) (*Babski et al., 2019*). MNB is the only unpaired neuroblast (*Wheeler et al., 2009*). It produces up to three ventral unpaired midline motor neurons before generating up to six GABAergic unpaired midline interneurons (*Figure 10A*; *Schmid et al., 1999*). In the A1 segment of the connectome, there are six unpaired Ladder interneurons with neurites that enter the neuropil in a single tight bundle (*Figure 10B*). We confirmed Ladders are progeny of MNB by crossing a fly line expressing membrane-GFP in one Ladder to three different fly lines, each expressing nls RFP in the MNB lineage. In all cases, Ladder and MNB lineage markers are found in the same cell (*Figure 10C–E*). In addition, we used distance analysis to evaluate input connectivity between Ladders in the A1 segment (*Supplementary file 3*) and find statistically significant similarities (*Figure 10F*, *Figure 10—source data 1–2*). We conclude that Ladders are a late-born temporal cohort from MNB. Like other temporal cohorts, we find that the Ladder temporal cohort border correlates with sharp transitions in synaptic connectivity.

## A08 neurons that synapse with early-born ELs in A1 are early-born ELs from other segments

We noticed that at least some A08 interneurons that synapses with early-born ELs in A1 resembled early-born ELs in terms of morphology. Therefore, we tested the idea that these A08 interneurons could be NB3-3 progeny in segments other than A1. To do so, we used the multi-color flip out (MCFO) system to highlight the morphology of individual neurons within the expression domain of a GAL4 line (*Nern et al., 2015*). For GAL4 lines, we used either *early-born EL-GAL4* or *late-born EL-GAL4* (*Wreden et al., 2017*). We compared early-born and late-born EL morphologies to the A08 interneurons that formed synapses with early-born EL in A1. We found that a majority were early-born ELs from segments other than A1, including anterior (i.e., thorax) and posterior (i.e., A2) segments (*Figure 11A–F*, *Supplementary file 1*). This reveals inter-segmental synaptic connectivity between early-born ELs (*Figure 11G*). Functionally, ELs are inter-segmentally coordinated (*Heckscher et al., 2015*). The observed inter-segmental synaptic connectivity could help explain this observation.

Because we found inter-segmental synaptic connectivity between early-born EL temporal cohorts, we wondered if this occurred between other temporal cohorts. We found inter-segmental connections between Basins and Ladders, but not late-born ELs (*Figure 11G*, *Supplementary file 1*, *Supplementary file 3*). In addition, in A1, we found intra-segmental (within-segment) synaptic connections between early-born ELs, Basins, Ladders, but not late-born ELs (*Figure 11G*, *Supplementary file 1*, *Supplementary file 3*). One commonality between early-born, Basins, Ladders, but not late-born ELs, is that they receive synapses from CHOs. This raises the possibility that inter- and intra-segmental synaptic connectivity between these temporal cohort is a feature important for the processing of vibrational stimuli. Further, these data reveal differences in connectivity between temporal cohorts, hinting at complexity in mechanisms driving their wiring.

In summary, we find that Basins neurons are a mid-to-late-born temporal cohort from NB3-5. Ladders are late-born temporal cohort from MNB. And A08 interneurons are early-born ELs from segment other than A1. This supports the hypothesis that in the *Drosophila* larval nerve cord circuits are assembled by preferential wiring between temporal cohorts. Further, we note that temporal cohorts that form synapses on early-born ELs come from 3 of the 30 neuroblast in the nerve cord. This finding rules out the following two models: (1) that morphologically similar neurons from different lineages

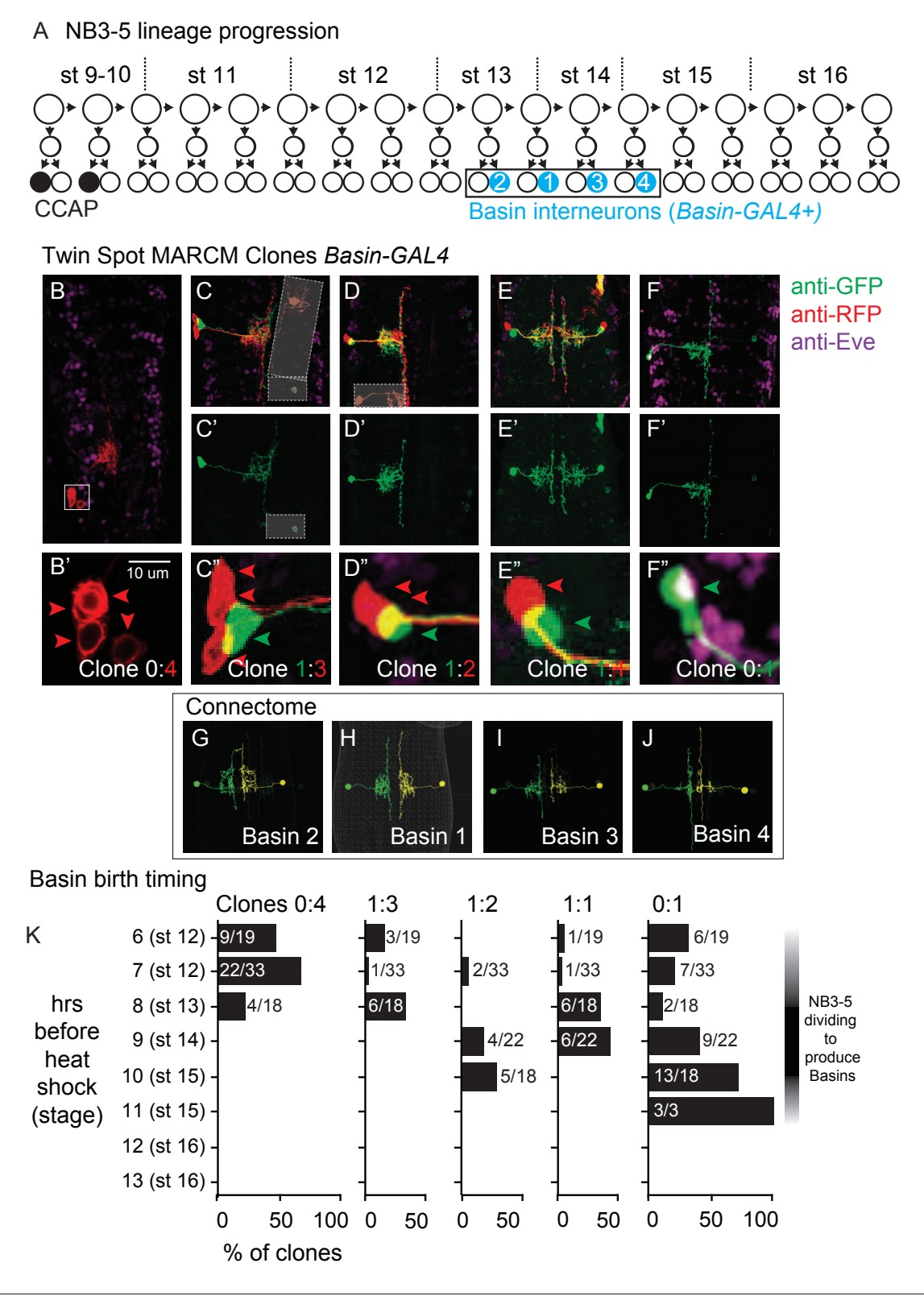

**Figure 8.** Birth time and birth order of Basin interneurons using twin-spot mosaic analysis with a repressible cell marker (ts-MARCM). (**A**) Illustration of NB3-5 lineage progression. Each circle represents one cell, and each arrow represents a cell division. The X-axis represents developmental time, and dashes represent approximate positions of embryonic stages (e.g., st16). NB3-5 generates two CCAP(+) neurons (black) followed by a series of other neurons. Basins (cyan) are born a middle-to-late window. (**B–J**) Images of Basin ts-MARCM clones and Basins in the connectome. Examples of

*Figure 8 continued on next page*

*Figure 8 continued*

ts-MARCM clones are shown in (**B**–**F'**). Eve staining serves as a counterstain to visualize the nerve cord architecture. RFP and GFP show ts-MARCM Basin clones. Genotype in ***Supplementary file 4***. (**B**) shows an example of a clone produced by an early, brief heat shock (see 'Materials and methods'). In this nerve cord, only one clone is present, and in that clone, all four Basins are labeled. (**B'**) corresponds to the boxed region in (**B**). Red arrowheads point to each cell body (two are stacked on each other). (**C**–**F"**) show examples of other clone types. (**C**–**F**) shows two color labeling. (**C'**–**F'**) show morphology of the singly labeled Basin. (**C"**–**F"**) show higher magnification of the cell bodies with arrowheads pointing to labeled cells. Clone types indicated at the bottom of the panel. In (**C**), boxes with white dashes have been placed over 'off target' clones for visual clarity. (**G**–**J**) show left–right pairs of Basin interneurons in segment A1 of the connectome. All images shown in dorsal view with anterior up. (**K**) Quantification of types of clones produced by variously timed heat shocks. The clone type is displayed at the top of each graph. Y-axis for each graph shows the various times after egg collection until heat shock was applied. The X-axis for each graph represents the percentage of clones of that clone type that were produced by a given heat shock protocol. Numbers are the total number of clones of a clone type over the total number of clones scored for that time point.

synapse onto early-born ELs, and (2) that multiple temporal cohorts from a single neuronal stem cell synapse onto early-born ELs. Instead, it shows that neurons from a limited number of neuronal stem cell lineages selectively synapse onto early-born ELs.

## Presynaptic interneurons are born after their postsynaptic interneuron partners

To more fully understand assembly of this feed-forward circuit, we needed to understand the relative birth timings of early-born ELs, Basin, and Ladder interneurons. One model for motor circuit development suggests that early-born neurons wire with each other to generate circuits driving fast movements, whereas later-born neurons wire with each other to generate circuits driving refined movement (i.e., early-to-early and late-to-late model) (***Fetcho and McLean, 2010***; ***Ampatzis et al., 2014***; ***Mark et al., 2021***). Furthermore, we found that early-born ELs from different segments synapse with each other, raising the possibility that an early-to-early model could also explain the observed pattern of wiring. However, there is suggestive evidence that an exclusive early-to-early model cannot explain the full complexity of wiring patterns seen in motor circuits (***Kishore et al., 2014***; ***Menelaou et al., 2014***; ***Song et al., 2018***). Thus, here, we test the hypothesis that early-born ELs wire with neurons from other lineages, which are born at the same time as early-born ELs. Note that in the experiments described above we determined the birth order of neurons within a given lineage, but had no information about the relative birth timing of neurons between different lineages.

To determine the relative birth timing of early-born ELs, Basins, and Ladders, we started by asking when during neurogenesis each EL could be generated. We used ts-MARCM with *EL-GAL4*. We supplied heat shocks at different time points and scored the identity of singly labeled neurons. Heat shock supplied after 7 hr of development labeled a mixture of early-born and late-born ELs, whereas heat shock supplied after 9 hr labeled exclusively late-born ELs (***Figure 12A and B***). Thus, heat shocks after ~9 hr failed to label early-born ELs. Next, we asked when during neurogenesis the Basins and Ladders are generated. We used ts-MARCM with GAL4 lines that labeled individual Basin or Ladder neurons (Basin-1 or Ladder-D) (***Figure 12C and D***). We supplied heat shocks at different time points and scored for the proportion of CNS in which clones were induced. In both cases, heat shocks provided at 11 hr of development, or later, failed to label these neurons. Whereas heat shocks provided at 10 hr of development, or earlier, resulted in labeling (***Figure 12E***). Thus, heat shocks at ~10 hr are sufficient to label Basins and Ladders. Taken together these data strongly suggest that Basins and Ladders are generated after the early-born ELs. Notably, this is inconsistent with the model in the field.

Because our data were inconsistent with our expectation, we performed an additional set of experiments to confirm our findings. Specifically, we performed dual ts-MARCM using both *Basin-GAL4* and *EL-GAL4* in the same larva. This approach allows for direct comparison of neurons generated at a given time in a single animal. We provided a series of heat shocks and scored the resulting clones as follows: EL clones were scored for neuron identity (e.g., A08j2), and this was used to assign ELs as early-born or late-born. Basin clones were scored for clone type (e.g., 1:3 or 1 neuron labeled in one color, 3 neurons in another color), and this was used to determine the time window of Basin production. Specifically, if all Basins were labeled in a single color (0:4 clone), the heat shock was provided before the neuroblast began to divide to produce Basins. Alternatively, if Basins were singly labeled (e.g., 1:3, 1:2, etc.), the heat shock was provided to cells as they were dividing to produce Basins.

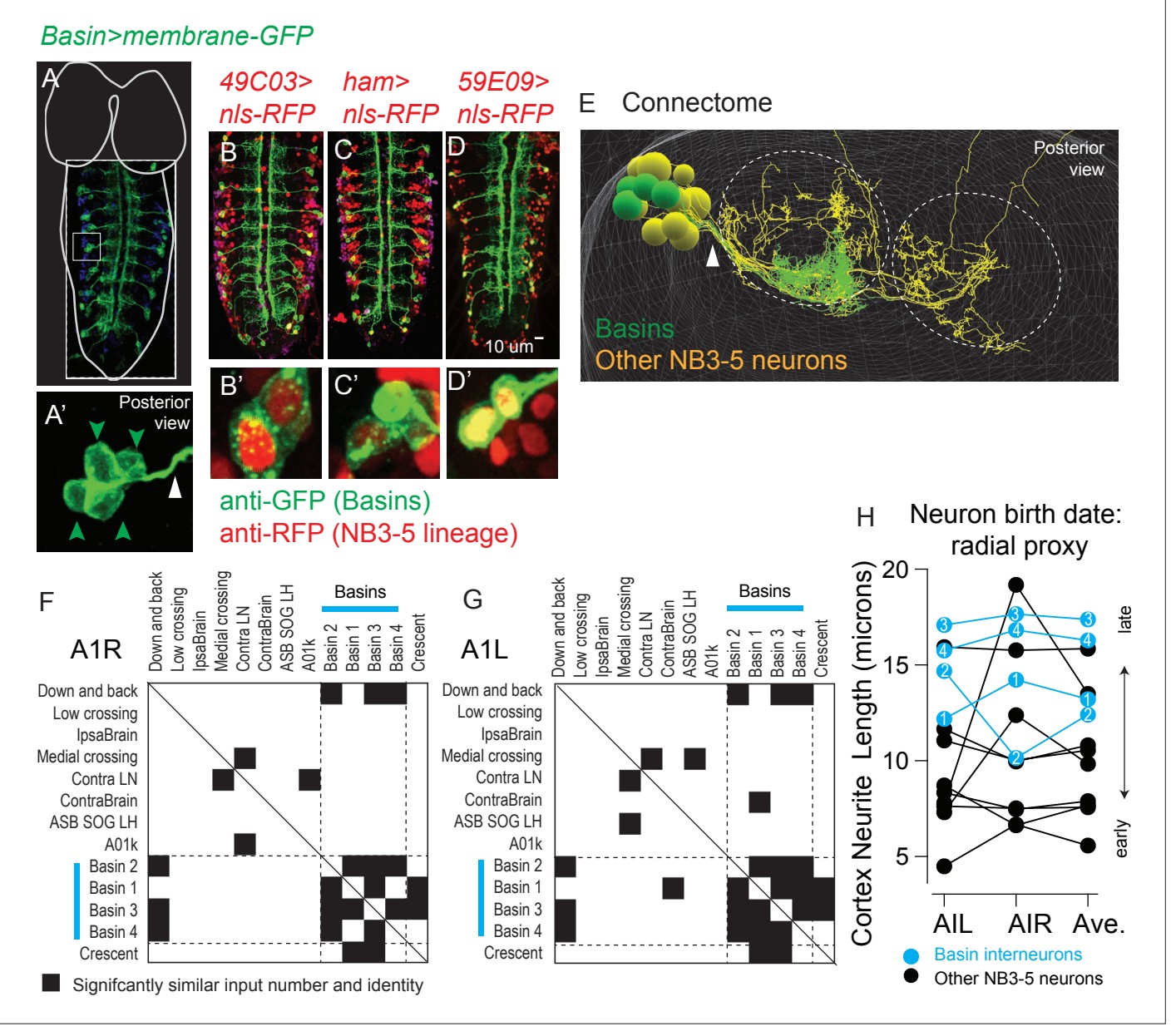

**Figure 9.** Basins are a middle-to-late-born temporal cohort in the NB3-5 lineage. (**A–E**) Images of Basin interneurons and other lineage-related neurons. (**A**) The larval central nervous system (CNS) with Basins neurons expressing membrane GFP is shown in dorsal view with anterior up. For context, the outline of the nerve cord and brain lobes is shown in white. Dashed box outlines the image. The small light box region is shown in (**A'**). The green arrowheads point to Basin cell bodies, which are clustered. The white arrow points to bundle of Basin neurites. (**B–D'**) Larval nerve cords show co-expression of a Basin marker (membrane GFP) and various NB3-5 progeny markers (nuclear localized RFP). Genotypes in *Supplementary file 4*. (**E**) An image of NB3-5 progeny in the connectome is shown. The neuropil is outlined by dashed circles. Basin neurons are in green, and other neurons in the NB3-5 lineage are in yellow. Arrowhead points to the bundle containing Basins. (**F–H**) Quantification of Basin features using connectome data. (**F, G**) To quantify the similarity in wiring between neurons in the NB3-5 lineage, normalized (nonbinary) distance plots were generated. Right and left NB3-5A1 neurons are arranged in approximate order of their birth (based on average cortex neurite length) with Basins indicated by cyan bars. Row–column pairs with scores significantly smaller than shuffled controls are in black (p<0.05). The distance analysis plot is symmetric (solid line for the diagonal). Dashed lines are placed at the border between Basins and non-Basins. (**H**) The approximate birth order of Basins within the NB3-5 lineage was determined using cortex neurite length as a proxy. The Y-axis plots cortex neurite lengths for NB3-5 neurons in segment A1L, A1R, and their average. Compared to other neurons in the NB3-5 lineage, Basins are born near the end of the lineage.

The online version of this article includes the following source data for figure 9:

**Source data 1.** NB3-5 input neuron information.

**Source data 2.** Similarity z-score for NB3-5 A1R.

*Figure 9 continued on next page*

*Figure 9 continued*

**Source data 3.** Similarity z-score for NB3-5 A1L.

**Source data 4.** NB3-5 cortex neurite length.

Within our dataset, three animals had labeled early-born ELs. In all of these animals, Basins were labeled in a single color, consistent with the idea that the neuroblast had not yet begun generating Basins (*Figure 12F*). In 17 animals, we found singly labeled Basin neurons, suggesting that Basins were being generated. In 16 of these 17 animals, late-born ELs were labeled (*Figure 12F*). Together, these data provide strong evidence that early-born ELs are born earlier than the Basin interneurons.

In summary, we find that between neurons in the feed-forward circuit, early-born EL are born before Basin and Ladder interneurons (*Figure 12G*). More generally, this demonstrates that circuit outputs from one lineage are born before circuit inputs from other lineages. Further, this rules out an exclusive early-to-early hypothesis that states that neurons born at the same time are circuit partners.

## Other neurons that are highly connected to early-born ELs come from multiple different lineages and most are born after early-born ELs

In the *Drosophila* larval nerve cord, individual neurons make different numbers of synapses onto their various downstream partners. In this section, we focus on neurons that are highly connected to early-born ELs, but not in the three lineages previously discussed. We considered a neuron to be highly connected if it has 10 or more synapses onto early-born ELs (*Supplementary file 6*). We choose this criterion for highly connected neurons, first, because a majority of synapses onto early-born ELs are made by these neurons, and second, because these neurons are likely to be important drivers of the activity of early-born ELs. In the sections above, we focused on Ladder, Basin, and Early-born EL interneurons, which, together with CHOs, account for the majority of neurons highly connected to early-born EL in A1 (*Figure 13A*, *Figure 13—source data 1*, *Supplementary file 6*). From these data, we learned two 'rules.' First, interneurons in the feed-forward circuit come from temporal cohorts in three lineages. Second, output interneurons from one lineage are born before input interneurons from other lineages. Additionally, there are six other highly connected neurons (*Figure 13A–C*). Here, we characterize the developmental origins of these other neurons to understand the extent to which the rules apply.

First, we wanted to understand the extent to which the six other highly connected interneurons could be lineage-related. To assign lineage identity, we examined their cell body position and neurite fasciculation pattern. The six other neurons likely come from six different lineages (*Figure 13B and C*). Of these, one is NB3-5, which generates Basin neurons. This means that NB3-5 neurons contribute more synapses onto early-born ELs than we had previously appreciated. We conclude that together NB3-5, MNB, and NB3-3 generate a majority (75%) of interneurons that are highly connected to early-born ELs. However, the relationship is not exclusive, and interneurons from other lineages do also synapse multiple times onto early-born ELs.

Next, we wanted to understand the extent to which the six other highly connected interneurons could be born after early-born ELs. To do so, we examined the cell body position of each along the medio-lateral axis (*Figure 13B'*). Of the six, one had a medial cell body close to the neuropil, suggesting that it was early-born within its lineage; four had cell body positions on or close to the lateral edge of the CNS, suggesting that they were late-born with their respective lineages; and one had an intermediate cell body position that is difficult to interpret (*Figure 13C*). These data, along with our previous birth dating from Basins, Ladders, and ELs, suggest that 70% of highly connected interneurons are born after early-born ELs. We conclude that the large majority of highly connected neurons in this circuit are born after the circuit output neurons (early-born ELs), but that there are exceptions to this 'rule'.

Finally, for one of the other highly connected other interneurons, we wanted to experimentally validate our lineage assignment and more directly determine its birth time relative to early-born ELs. We chose to focus on neuron *A03 upstream of A08m* because it is inferred to be progeny of NB7-1, and the lineage NB7-1 is extremely well-characterized. NB7-1 delaminates early during neurogenesis (*Broadus et al., 1995*). The first six divisions of NB7-1 generate motor neurons and undifferentiated motor neuron siblings (*Seroka and Doe, 2019*; *Lacin et al., 2009*). Then, NB7-1 divides

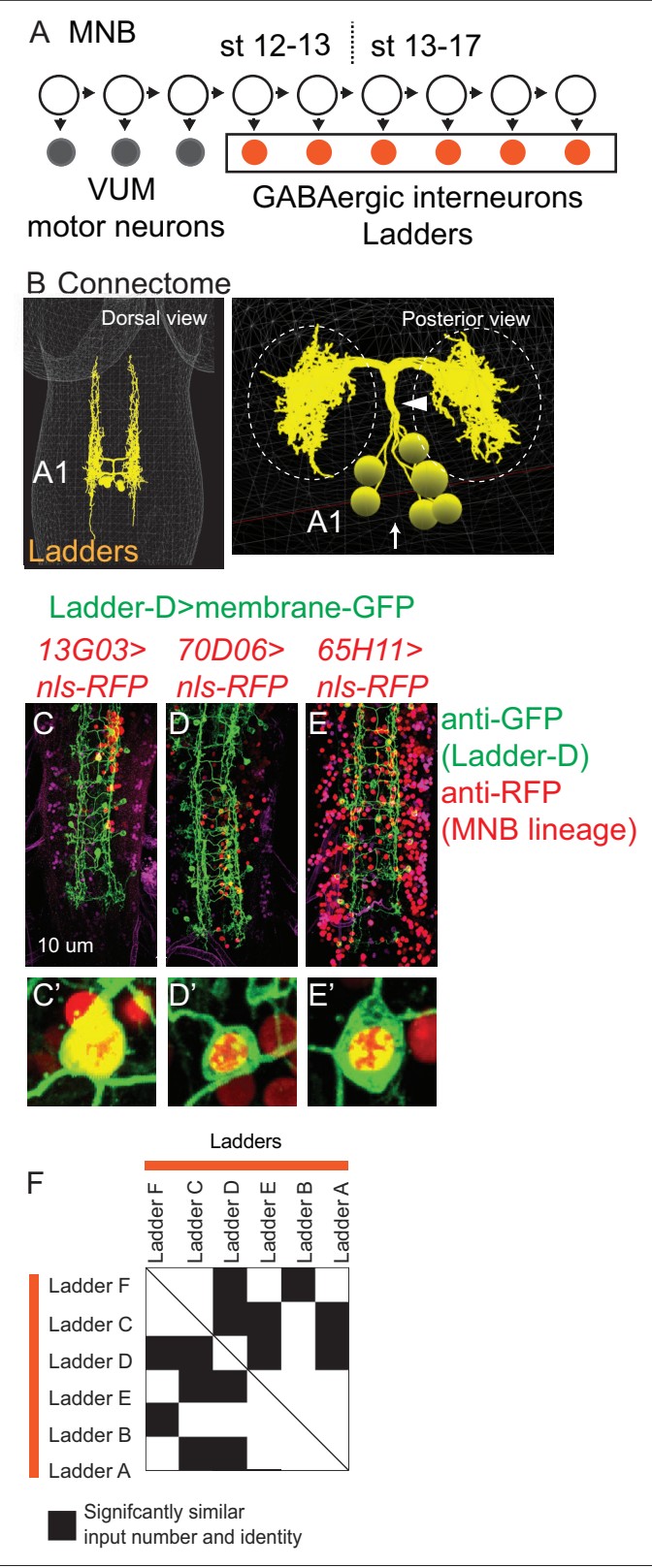

**Figure 10.** Ladders are a temporal cohort from midline neuroblast (MNB). (**A**) Illustration of MNB lineage progression. Each circle represents one cell, and arrows represent cell division. The X-axis represents developmental time, and dashes represent approximate positions of different embryonic stages (e.g., st16). MNB generates up to three, early-born ventral unpaired midline motor neurons (gray) followed by a series of GABAergic

*Figure 10 continued*

interneurons (orange). These interneurons are Ladders. (**B–E**) Image of Ladder interneurons in the connectome and of co-expression of Ladder-D interneuron and MNB lineage markers. (**B**) Ladders in segment A1 of the connectome shown in dorsal view and side views. Ladders form a bundle (arrowhead) before entering the neuropil (dashed circles). Midline shown with arrow. (**C–E'**) First-instar larval central nervous systems (CNSs) are shown in ventral view with anterior up. In insets, notice co-expression of Ladder-D membrane marker (green), with nuclear localized MNB lineage maker (red). Genotypes in *Supplementary file 4*. (**F**) Quantification of statistical similarities in Ladder synaptic inputs. Plot show normalized (nonbinary) Euclidean distance similarity between pairs of Ladders neurons. Ladders are arranged in approximate order of their birth based on cortex neurite length. Row–column pairs with scores significantly (p<0.05) smaller than shuffled controls preserving the inputs number and magnitude are in black. Euclidean distance plot is symmetric (solid line for the diagonal).

The online version of this article includes the following source data for figure 10:

**Source data 1.** Ladder input neuron information.

**Source data 2.** Similarity z-scores for Ladder neurons.

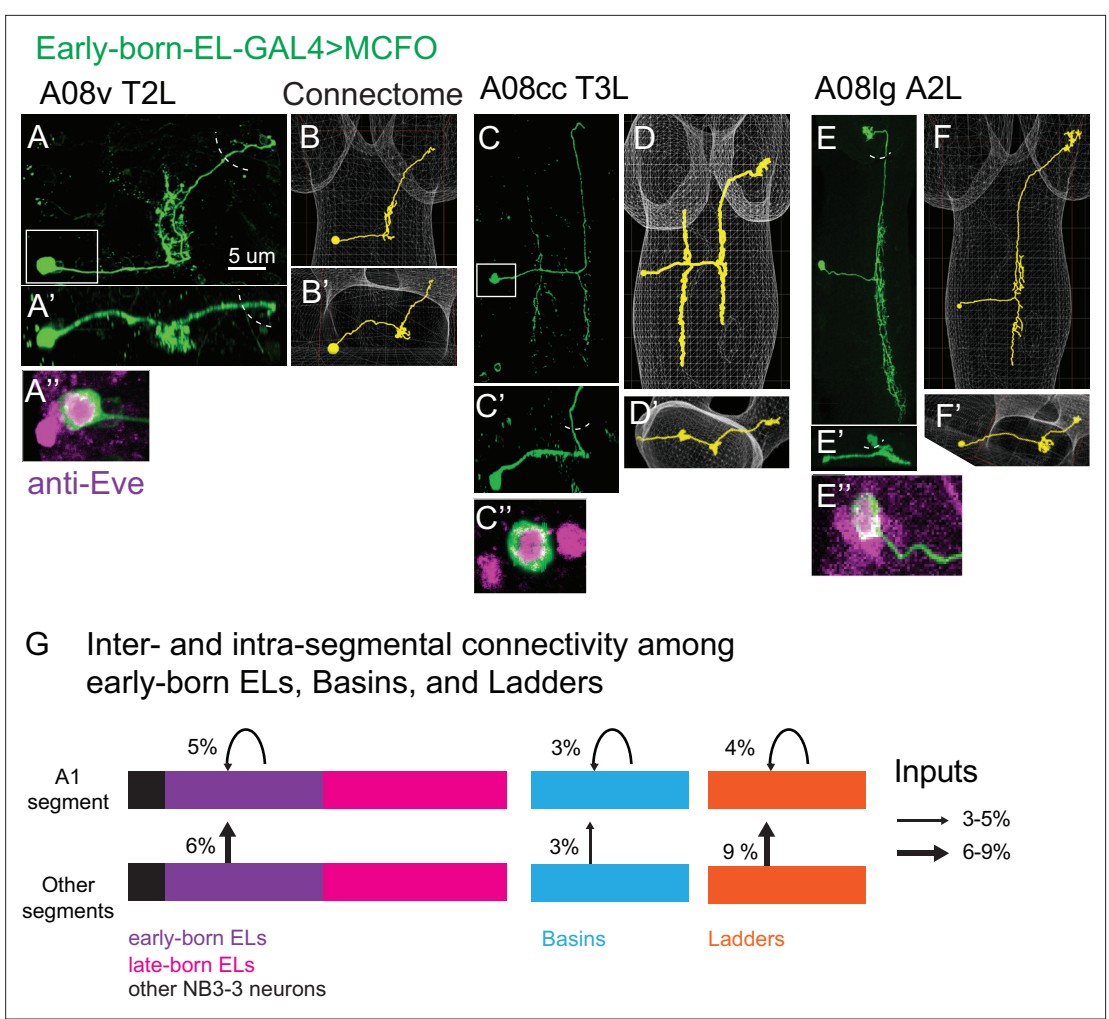

**Figure 11.** A08 interneurons that synapse onto early-born ELs in A1 are early-born ELs from segments other than A1. (**A–F**) Images of early-born EL interneuron multi-color flip out (MCFO) clones along with matching A08 interneurons in the connectome. For each figure panel, the main image (e.g., **A**) shows neuronal membrane labeling in a dorsal view with anterior up. The same cell is shown in posterior view (e.g., **A'**). Boxed areas are magnified to show soma co-labeling with Eve, which demonstrates it is an EL (e.g., **A"**). Dashed semi-circles show approximate position of central brain lobes. Confocal images (e.g., **A**) are shown adjacent to the matching neuron in connectome (e.g., **B**). Genotype in *Supplementary file 4*. (**G**) Illustration of inter-segmental and intra-segmental connectivity between early-born ELs, Basins, and Ladders, but not late-born ELs. Bars represent groups neurons, with black being non-ELs, purple early-born ELs, magenta late-born ELs, cyan Basins, and orange Ladders.

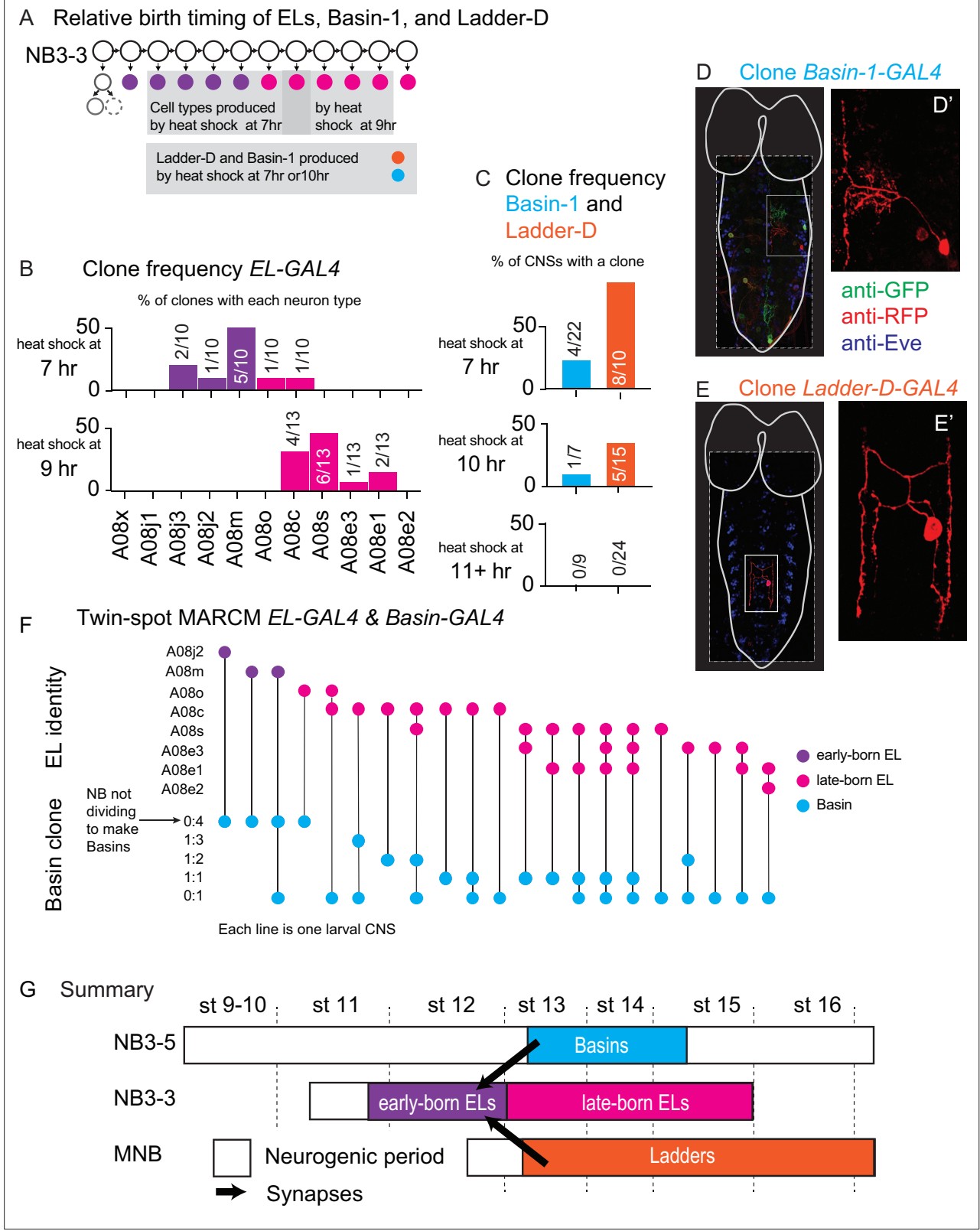

**Figure 12.** Basin and Ladder interneurons are born after early-born EL interneurons. (**A**) Illustration of neurons labeled by various heat shock protocols. At top, NB3-3 lineage progression is shown. Each circle represents one cell, and each arrow represents a cell division. Early-born ELs (purple) are labeled during a heat shock at 7 hr after egg collection (top-left gray box). Most late-born ELs (magenta) are labeled during a heat shock at 9 hr after egg collection (top-right gray box). Heat shocks provided at both 7 and 10 hr after egg collection are sufficient to label Basin-1 and Ladder-D (bottom

*Figure 12 continued*

gray box). (**B, C**) Quantification of cell types labeled by various heat shock protocols. (**B, C**) For each graph, the X-axis represents the type of neuron produced. For ELs (in **B**) neuron names are presented in birth order. The Y-axis represents how often a given neuron was generated by that protocol with early-born ELs in purple, late-born ELs in magenta, Basin-1 in cyan and Ladder-D in orange. In (**B**), numbers in each bar represent the number of labeled neurons of each type over the total number of neurons scored. In (**C**), numbers in each bar represent the number of central nervous systems (CNSs) with a labeled neuron over the total number of animal CNSs scored. Genotype in *Supplementary file 4*. (**D, E**) Images of singly labeled Basins or Ladders. Larval CNSs are shown in ventral view with anterior up. For context, the outline of the nerve cord and brain lobes is shown in white. Dashed box outlines the image. The small light box shows region in (**D', E'**). (**D'**) shows a singly labeled Basin-1. (**E'**) shows a singly labeled Ladder-D. Genotype in *Supplementary file 4*. (**F**) Quantification of ELs and Basins labeled in a single CNS. The Y-axis is divided into two sections. The top shows EL identity and describes a specific A1 EL interneuron type (e.g., A08j2) listed in order of their birth. The bottom shows Basin clone type and refers to the pattern of labeling of Basin neurons. For example, 0:4 means four neurons were labeled in one color and none another. Each column (X-axis) represents a different larval CNS, with a dot indicating the type of neuron labeling observed. A line connecting the two clone types to help visualize the pairs. In some CNSs, more than one type of Basin clone was produced, and so multiple dots are present. (**G**) Illustration of relative birth timing of Basins, Ladders, and early-born EL interneurons. Each row represents neurons generated from a different neuroblast over time. The X-axis represents developmental time, and dashes represent approximate positions of different embryonic stages (e.g., st16). Early-born ELs (magenta) get input from neurons born after they are born.

approximately 14 more times, generating interneurons (*Meng et al., 2019*), one of which is thought to be neuron *A083 upstream of A08m* (*Mark et al., 2021*). We labeled neuron *A03 upstream of A08m* using a *NB7-1-GAL4* driver line to create MCFO clones, which validates the lineage assignment (*Figure 14A and B*). Then, in the connectome, we measured the cortex neurite length of neuron *A03 upstream of A08m* and the other interneurons within the same hemilineage as a proxy for birth order (*Figure 14C and D*, *Figure 14—source data 1*). At the earliest, neuron *A03 upstream of A08m* is the sixth-born *interneuron*, and because there are six divisions of NB7-1 before interneurons are made, neuron *A03 upstream of A08m* must be born on or after the 12th division (*Figure 14E*). To compare the birth timing of neuron *A03 upstream of A08m* to the birth timing of early-born ELs, we present the following logic. NB3-3 and NB7-1 are generated at ~400 min and ~230 min after egg laying, respectively (*Doe, 1992*). Neuroblasts divide every 45 min. And so, NB3-3 is four divisions behind of NB7-1 ([400 min –230 min]/45 divisions/minute = 3.8 divisions). Early-born ELs are made during the second to sixth divisions of NB3-3, which corresponds to the fifth to ninth divisions of NB7-1 (*Figure 14E*). Because NB7-1 produces neuron *A03 upstream of A08m* no earlier than the 12 divisions, we conclude that it must be born after the early-born ELs. In addition, we noticed that three other interneurons in the same hemilineage as 'A03 upstream of A08m' synapse onto early-born ELs, albeit less strongly. The cortex neurite length of these neurons shows that they largely have birth times adjacent to neuron *A03 upstream of A08m* (*Figure 14D*), which is consistent with the idea that these neurons are a temporal cohort. Using the same logic as described earlier, we find that all the NB7-1 interneurons that synapse onto early-born ELs are born after the early-born ELs themselves (*Figure 14E*). Thus, analysis of NB7-1 shows that neurons from a third lineage follows the patterns identified for Ladder and Basin temporal cohorts. Furthermore, these data show that additional temporal cohorts have neurons that synapse onto early-born ELs, but not as extensively as neurons in the Ladder, Basin, or early-born EL temporal cohorts.

## Discussion

In this study, we addressed two questions about the stem cell-based assembly of neuronal circuits. First, what is the relationship between neuronal birth order within a lineage and patterns of synaptic connectivity at the single-neuron level? Second, how do neurons from different lineages wire with each other? We characterized the birth order, morphology, and input connectivity of all neurons in the NB3-3A1L/R lineage at single-neuron and single-synapse resolution (*Figures 1–5 and Supplementary file 1*). We identified a feed-forward circuit that encodes the onset of vibrational stimuli (*Figure 7*). And, for a majority of nerve cord interneurons within this circuit, we identified their stem cell parent, birth order within their lineage, and birth timing relative to each other (*Figures 8–14*, *Supplementary files 2-4*). Together, this identifies four temporal cohorts, all of which have sharp connectivity boundaries. For most, but not all, there is inter-segmental connectivity between segmentally homologous temporal cohorts (e.g., early-born ELs in other segments connect to early-born ELs in A1, *Figure 11G*). Further, neurons of different temporal cohorts from different lineages assemble

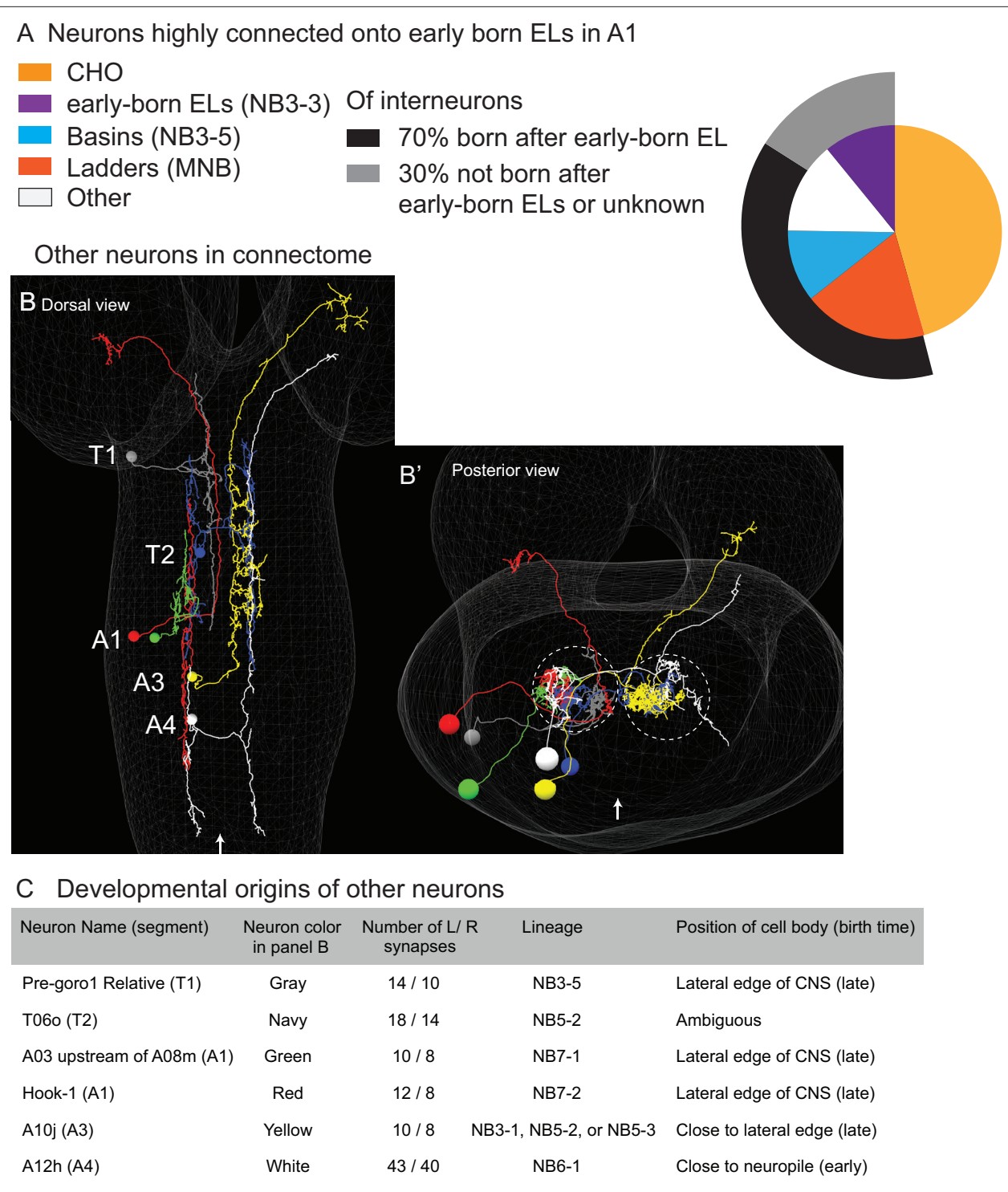

**Figure 13.** Other interneurons that are highly connected onto early-born ELs in A1 come from multiple lineages, and a majority are born after early-born ELs. (**A**) Quantification of neurons highly connected to early-born ELs. The pie chart displays highly connected inputs onto early-born ELs as a proportion of total number (70) highly connected neurons. Highly connected is 10 or more synapses from an individual neuron onto early-born ELs in A1. Each slice is the percentage of neurons from a given lineage. (**B**) Images of highly connected neurons in the connectome. (**B, B'**) For each image, a faint white mesh shows the outline of central nervous system (CNS) volume. (**B**) is a dorsal view with anterior up, and (**B'**) shows the CNS in a posterior view with dorsal up. Segment names are shown in (**B**). Midline is shown as an arrow in (**B**) and (**B'**). In (**B'**), circles outline the neuropil. The left member or each left–right pair of highly connected 'other' neuros are shown, each in a different color. Color code as in (**C**). (**C**) Table summarizing developmental origins of other highly connected neurons.

*Figure 13 continued on next page*

*Figure 13 continued*

The online version of this article includes the following source data for figure 13:

**Source data 1.** Summary of amount and timing of important input onto early-born ELs.

sequentially, with circuit output neurons born before circuit input neurons. Additionally, this study provides new tools to study stem cell-based assembly of a fundamental circuit motif.

## New resources for analysis of connectome data

The *Drosophila* larval connectome is a resource that can be used to understand circuit assembly (*Meng and Heckscher, 2021*). However, because the connectome is an anatomical dataset, a major challenge is to develop approaches that connect anatomy to development. In this article, we develop several approaches. For example, we optimized ts-MARCM for use in *Drosophila* embryos (*Figure 1*). ts-MARCM is the gold standard for determining birth order in *Drosophila* (*Yu et al., 2009b*). However, for technical reasons, ts-MARCM has been used only in adults. We discovered ts-MARCM clones can be robustly generated in early stage larvae with the addition of an amplifying and immortalizing gene cassette (*Figure 1*). In addition, we independently validated several recently developed methods for inferring developmental origin based on anatomical features in the *Drosophila* larval connectome (*Mark et al., 2021*). Specifically, our validations include use of neurite bundles as a proxy for lineage-relatedness (*Figures 3B, C, 9A, E, 10B and 14C*) and use of cortex neurite length as a rough proxy for birth order within a lineage (*Figures 3E, 9H and 14B*). Finally, we adapted network science methods (distance analysis) to characterize the patterns of connectivity in connectome data (*Figures 5, 9 and 10*). These approaches should be useful for *Drosophila* neurobiologists and beyond.

We also developed NB3-3A1L/R as the first entire lineage for which birth order, morphology, and input connectivity is known at single neuron and synapse precision (*Figures 2–4*, *Supplementary file 1*). One reason this is important is because NB3-3 has been extensively studied in embryos and much is known about molecular marker expression of NB3-3 progeny at the single-neuron level (*Tsuji et al., 2008*; *Wreden et al., 2017*; *Baumgardt et al., 2014*). Because our dataset achieves cellular resolution, good guesses about the embryonic molecular–larval morphological pairings can be made using single-neuron birth order as a point of cross-reference. Such integrated data generates detailed and testable predictions. For example, our data predict that Castor expression in the late-born ELs promotes projection neuron morphology. Additionally, here, we generated a new *NB3-3-GAL4* line that can be used to manipulate gene expression in NB3-3 (Figure S1). Thus, NB3-3A1L/R is a model lineage in which transcription factor expression in neuronal stem cells can be linked to circuit assembly and tested for function, and we have generated tools that will enable hypothesis testing in the future.

## Early-born ELs are embedded in a potentially conserved, feed-forward circuit motif

A circuit motif is a pattern of synaptic connections between a set of specific neuron types that can be found across brain areas and across species (*Braganza and Beck, 2018*). Circuit motifs have been suggested to represent the physical substrates of 'computational primitives' (*Marcus et al., 2014*). There a are small number of fundamental, recurring circuit motifs (e.g., feed-forward, feedback, lateral inhibition, etc.; *Luo, 2021*). And so, a new conceptual approach in this article is to ask how are specific circuit motifs assembled during development. In this study, we identified a new feed-forward circuit motif (*Figure 7*). Generally, feed-forward motifs are characterized as a pattern of connectivity in which one neuron (or neuron type) provides both direct and indirect input onto a second neuron (or neuron type). Feed-forward circuit motifs are common, found in many animals (e.g., nematodes, insects, mouse) and in many brain regions (e.g., somatosensory systems, olfactory systems, neocortex) (*Schafer, 2016*; *Anton and Homberg, 1999*; *Harris and Shepherd, 2015*). Thus, feed-forward motifs are fundamental to neural signal processing. Feed-forward motifs can be further subdivided into feed-forward inhibitory or feed-forward excitatory circuit motifs, depending on the transmitter types of the neurons involved(*Luo, 2021*). Notably, anatomically the pattern of synaptic connectivity between neurons is the same in either motif subtype, and so in this study we do not distinguish between the two.

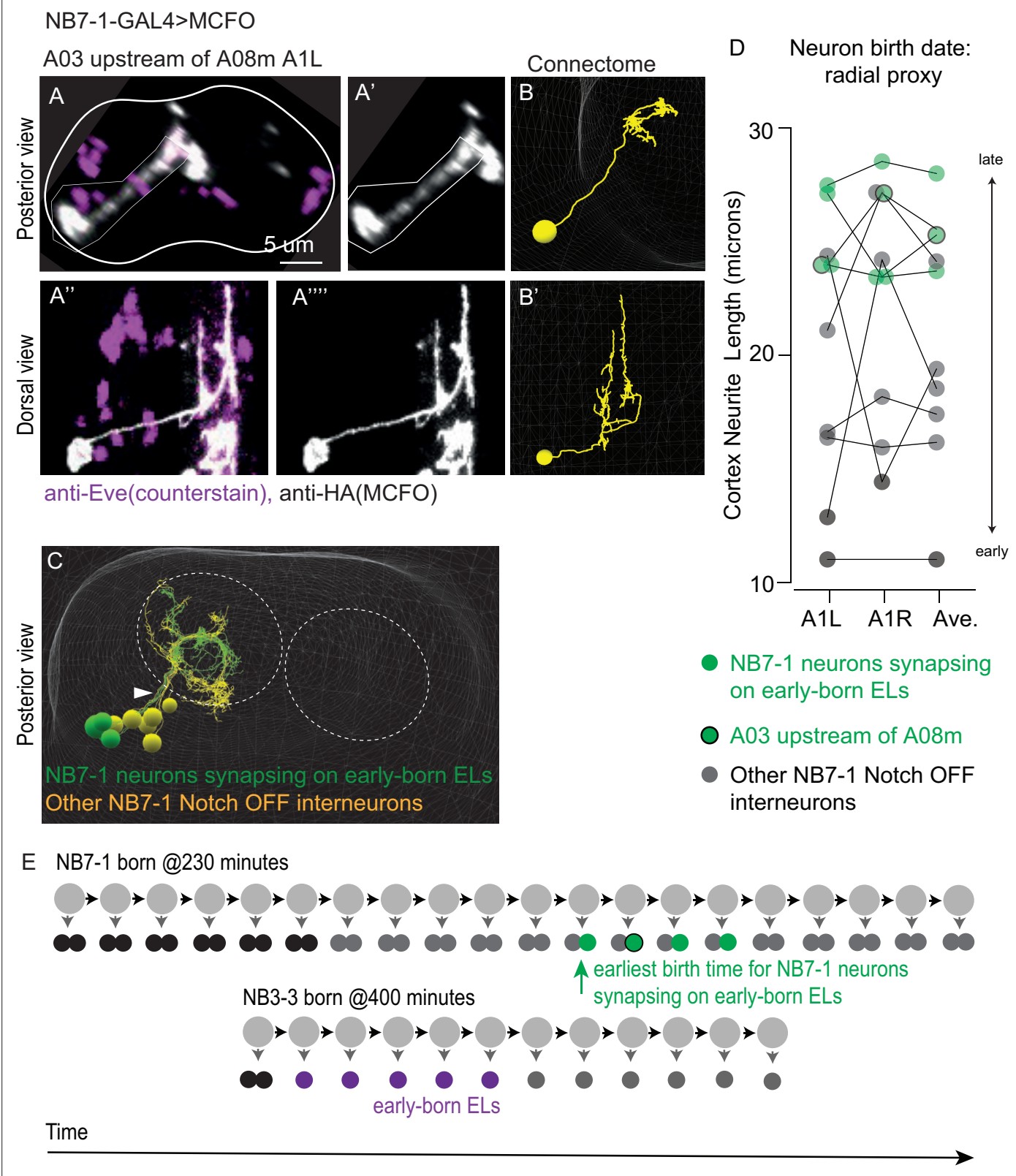

**Figure 14.** Interneurons from the NB7-1 Notch OFF hemilineage synapse onto early-born ELs and are born after early-born ELs. (**A–C**) Images of neuron *A03 upstream of A08m*, an interneuron from the Notch OFF NB7-1 hemilineage. (**A**) A confocal image shows a labeled neuron *A03 upstream of A08m*. This clone was generated using multi-color flip out driven by NB7-1-GAL4. In (**A**) and (**A"**), an anti-Eve counterstain is also shown. Image borders are noted and the outline of the central nervous system (CNS) is shown in (**A**). The confocal image is shown adjacent to the matching neuron in the

*Figure 14 continued on next page*

*Figure 14 continued*

connectome (**B, B'**). (**C**) An image of all interneurons in the NB7-1 Notch OFF hemilineage in segment A1L is shown. The neuropil is outlined by dashed circles. Neurons that synapse onto early-born ELs are in green, and other neurons in the hemilineage are in yellow. Arrowhead points to the lineage bundle. For genotype, see *Supplementary file 4*. (**D**) Quantification of neuron *A03 upstream of A08m* and other NB7-1 interneurons cortex neurite length. The birth order of A03 interneurons in the NB7-1 Notch OFF hemilineage was estimated using cortex neurite length as a proxy. The Y-axis plots cortex neurite lengths for interneurons in the NB7-1 Notch OFF hemilineage in segment A1L, A1R, and their average. Compared to other neurons in the hemilineage (black), NB7-1 interneurons that synapse onto early-born ELs (green) are born near the end of the lineage. (**E**) Illustration of NB7-1 and NB3-3 lineage progression showing relative birth timing of early-born ELs and NB7-1 interneurons that synapse onto them. Schematics show a summary of NB7-1 lineage progression compared to NB3-3 lineage progression. Each circle represents one cell with large circles as neuroblasts and smaller circles as neurons. Arrows represent cell division. The X-axis represents developmental time. During the first six divisions, NB7-1 generates motor neurons and undifferentiated siblings (black), and in the remaining divisions generates interneurons (gray and green). NB3-3 starts to divide approximately four divisions than NB7-1. Nonetheless, early-born ELs (magenta) are born before the NB7-1 neurons that synapse onto them (green).

The online version of this article includes the following source data for figure 14:

**Source data 1.** NB7-1 naming and cortex neurite length.

Here, we find that, in general, early-born ELs get direct excitatory synapses from CHOs and indirect (excitatory or inhibitory) input from chordotonals via Basins and Ladder interneurons (*Figure 7*). Notably, there are differences in connectivity patterns between early-born ELs. For example, although all early-born ELs get direct input from chordotonals and Ladders, A08j1-3s get left–right symmetrical inputs, whereas A08x and A08m get asymmetrical inputs (*Figure 7*). This could correspond to functional differences. For example, A08x and A08m may be involved in left–right asymmetrical signal detection. Teasing apart the diversity of computations performed by each individual EL interneurons is the domain of future studies.

Because we characterized the development of a feed-forward circuit in *Drosophila*, we have information that allows us to search for similar circuits in other insect species. For example, in *Drosophila*, ELs express the transcription factor, Even-skipped (Eve, *Figures 1–2*). In Locust and other insects, lateral interneurons also express Eve (*Bevan and Burrows, 2003*). In *Drosophila*, MNB generates H-shaped, inhibitory interneurons (Ladders), which get input from sound/vibration sensitive, CHOs (*Figures 7 and 11*, *Supplementary file 3*). In Locust, MNB generates H-shaped, inhibitory interneurons, which encode sound stimuli (*Thompson and Siegler, 1991*). Further, in both *Drosophila* and Locusts, MNB interneuron progeny express the transcription factor Engrailed (*Schmid et al., 1999*; *Kearney et al., 2004*; *Patel et al., 1989*). Thus, the neuronal components of the Ladder to early-born EL circuit are conserved, which raises the possibility of circuit-level conservation.

## Deepening our understanding of temporal cohorts

A major unanswered question in developmental neuroscience is how the mechanisms that generate neuronal diversity contribute to the formation of functional neuronal circuits (*Sagner and Briscoe, 2019*). Part of the answer lies in the observation that temporal cohorts are subunits of lineages, linked both to larval circuit anatomy/function as well as to the embryonic gene expression programs that generate neuronal diversity (*Meng et al., 2019*; *Meng et al., 2020*; *Seroka and Doe, 2019*). Thus far, temporal cohorts had been looked for and found in 8 out of 30 lineages in the nerve cord (*Wreden et al., 2017*; *Meng et al., 2019*; *Meng et al., 2020*; *Mark et al., 2021*). Here, we identify three additional temporal cohorts—Basins, Ladders, and Notch OFF NB7-1 interneurons—in three lineages—NB3-5, MNB, and NB7-1 (*Figures 8–10* and *Figure 14*), bringing the number to 11. This underscores the idea that temporal cohorts are common.

One open question about temporal cohorts was to what extent are temporal cohort borders associated with sharp changes in connectivity. Previous studies had identified temporal cohorts and linked them with function and connectivity (*Wreden et al., 2017*; *Meng et al., 2019*; *Meng et al., 2020*; *Mark et al., 2021*). However, these studies lacked the resolution to distinguish between a 'graded' or 'sharp' wiring transition models. We identified four temporal cohorts (early-born ELs, late-born ELs, Basins, Ladders) in three lineages (NB3-3, NB3-5, MNB), all of which have sharp changes in connectivity correlated with temporal cohort borders (*Figures 5 and 9–10*). For the Basin temporal cohort, we note that Basins have similar connectivity patterns with two additional NB3-5 progeny, 'Down and Back' and 'Crescent' neurons. Crescent is adjacent to Basins in birth order, whereas Down and Back is born much earlier. From this, we learn two things: (1) there can be significantly similar connectivity

between neurons in one hemilineage that have nonadjacent birth times. (2) There can be significantly similar connectivity between neurons with adjacent birth times, but of different morphological classes. Another example of morphological variants with similar connectivity are late-born ELs, which can be either local or projection neurons. This underscores the idea that temporal cohorts are subunits of hemilineages defined by birth within a tight time window, rather than defined by similar neuronal morphology per se. Further, we note that within a temporal cohort, sequentially born neurons are often, but not always, the most similar in terms of connectivity. For example, within the early-born EL temporal cohort, the fifth-born neuron (A08m) is more similar to the first-born neuron (A08x) than its temporal neighbor (A08j2), the fourth-born EL. Thus, our data, combined with previous studies, show how temporal cohorts are developmental units related to circuits both at the functional and anatomical levels.

A second open question about temporal cohorts was the extent to which they are copies of each other. One reason this is an interesting question relates to circuit evolution. For the evolution of gene function, a popular model is a 'duplicate and diverge' model. Similarly, temporal cohorts could be duplications within a lineage whose function could then diverge, thereby driving circuit evolution. Such an idea motivated us to ask to what extent are temporal cohort copies. For example, do early-born ELs process chordotonal stimuli in a manner identical to how late-born ELs process proprioceptive stimuli? Our data suggest a more complex picture. Early-born and late-born ELs differ in their connectivity patterns—including left–right and following pair similarities (*Figure 5B*) and inter-and intra-segmental connectivity patterns (*Figure 11G–H*), suggesting that early-born and late-born ELs are likely to process information differently. Independent of the evolutionary implications, our data reveal previously unknown diversity in the structure of connectivity between neurons within adjacent temporal cohorts, which may indicate a diversity in underlying circuit assembly mechanisms.

A third open question is what sets up the borders of temporal cohorts. Previous work used motor neuron temporal cohorts of NB7-1 and NB3-1 as a model to address this question (*Meng et al., 2019*; *Meng et al., 2020*). In these stem cells, mis-expression of temporal transcription factors, Hunchback, Pdm, and Castor, modulates the number of motor neurons in a temporal cohort, without changing the size of the lineage. Thus, temporal transcription factors are able to regulate motor neuron temporal cohort borders. But it remains unclear how temporal transcription factors do so. Two, not mutually exclusive possibilities are that they act as transcriptional co-factors to induce differential gene expression programs and/or that they act as pioneer factors to alter the chromatin landscape. For NB3-3, we note that during the 7th to 11th divisions, which generate late-born ELs, NB3-3 expresses Grainyhead (*Tsuji et al., 2008*), which raises the possibility that Grainyhead may define the late-born EL cohort. Testing this idea is an important future direction.

## A feed-forward circuit is assembled from a select set of temporal cohorts

It has been hypothesized that nerve cord circuits are assembled by preferential connectivity between distinct temporal cohorts (*Meng and Heckscher, 2021*). Our data provide experimental support for this hypothesis. Specifically, we find that interneurons from three temporal cohorts wire together to form a feed-forward circuit—early-born ELs from NB3-3, mid-to-late-born Basins from NB3-5, and late-born Ladders from MNB (*Figures 8–10*). Although a vast majority of both the total synaptic input and the strongly connected individual neurons come from these lineages, there are also neurons from other lineages that synapse on early-born ELs (*Figures 13–14*).

Notably, one other study provided limited supported for the hypothesis that *Drosophila* nerve cord circuits are assembled by preferential connectivity between distinct temporal cohorts (*Mark et al., 2021*). This study focused on the Jaam-to-late-born EL-to-Saaghi circuit (*Heckscher et al., 2015*). Jaams are later-born interneurons in the NB5-2, Notch OFF hemilineage, and Saaghis are later-born interneurons in the NB5-2 Notch ON hemilineage (*Mark et al., 2021*). These data raised several possibilities: (1) there could be global alignment between lineages (e.g., all NB3-3 neurons get synapses from NB5-2 neurons). (2) Notch ON/OFF pairs of neurons might be pre-/postsynaptic partners of neurons within a temporal cohort. (3) Birth order-matched temporal cohorts from different lineages might selectively wire together (e.g., early-to-early and late-to-late connectivity). However, our data demonstrate that none of these possibilities are generally true. Instead, we find a diversity in

the manner in which temporal cohorts associate. The one consistent theme is that a limited number of temporal cohorts highly interconnect.

## Circuit outputs from one lineage are born before circuit inputs from other lineages

For most of this discussion, we labeled neurons as 'early-born' and 'late-born.' These labels refer to the birth order of neurons within a lineage. However, these labels do not refer to the absolute time at which a neuron is born. This is because in the *Drosophila* nerve cord neuroblasts are generated over a large span of embryogenesis (*Broadus et al., 1995*). Thus, early-born neurons from one lineage can be generated at the same time as later-born neurons from a different lineage. Our study is unique in that it determined both neuronal birth order and birth time.

One of our unexpected findings is that circuit outputs in one lineage are most often born before circuit inputs from other lineages. Broadly speaking, nerve cord and spinal cord contain many local circuits, which output to the brain (e.g., circuits processing somatosensory stimuli) or which output to muscles (e.g., circuits that generate motor patterns). Here, early-born ELs are the outputs of a somatosensory processing circuit and transmit information to the brain. Early-born ELs are born before their local, nerve cord inputs—Ladders, Basins, NB7-1 Notch OFF interneurons (*Figures 12–14*). However, from a first principles perspective, we would expect the opposite. This is because 22 of 30 neuroblasts in the nerve cord are born before NB3-3, and many of them divide multiple times to produce neurons before NB3-3 begins to generate early-born neurons (*Doe, 1992*). Therefore, there should be more neurons born before early-born ELs compared to those born after early-born ELs. And so, by chance alone we would expect early-born ELs to get more input from neurons born earlier or at the same time.

What are the hypotheses about the importance of birth order of output versus input neurons? In *Drosophila*, birth order is linked to two things: (1) lineage-intrinsic factors such as dynamically changing programs of gene expression in the stem cell, and (2) lineage-extrinsic factors or the dynamic environmental context into which neurons are born. Intrinsic factors, or extrinsic factors, or both may be playing a role in the assembly of this feed-forward circuit. A potential intrinsic mechanism is that of a temporal transcription factor 'matching code,' in which early-born ELs, Ladders, and Basins would all be derived from the same temporal transcription factor window. For NB3-5 and MNB, temporal transcription factor expression is only partially characterized. But tantalizingly both early-born ELs and a subset of Ladders are born during a period in which their respective neuroblasts express the temporal transcription factor, Castor (*Tsuji et al., 2008*; *Kearney et al., 2004*). A potential extrinsic mechanism is that early-born EL dendrites may provide some type of signal that promotes later born neurons to synapse. There is evidence for such communication among *Drosophila* nerve cord neurons (*Valdes-Aleman et al., 2021*). Finally, it will be interesting to understand if sequential assembly is an absolute requirement, or if instead it facilitates rapid, efficient, or robust circuit assembly.

## In what circumstances does sequential addition of temporal cohorts from different lineages occur?

Studies of circuit assembly are still in their infancy. It is known that lineage–circuit relationships differ depending on circuit anatomy. But the converse is not known. That is, do all circuits of the same anatomy have a common lineage–circuit relationship? In the case of this study, we focused on a single feed-forward circuit in the *Drosophila* larval nerve cord. And the specific question it raises is: To what extent is sequential addition of temporal cohorts from different lineages the only mechanism used to assembly all feed-forward motifs? Currently, our answers are only partial and speculative. We note that there are so many connections made among neurons, it is unlikely that just one simple phenomenon that can explain the full complexity. The goals of the current research must be to identify rules and the circumstances where those rules apply. For example, our data rule out a 'strict' early-to-early, late-to-late model, meaning that this model alone cannot explain the wiring we observe in this circuit. And yet, an early-to-early model does apply to inter-segmental wiring among temporal cohorts of the same lineage. Further, this early-to-early wiring phenomenon occurs alongside a sequential addition phenomenon. We speculate there could be additional phenomena underlying assembly of this simple motif. For example, beyond local interneurons, we did not determine the birth date of sensory neurons, which provide the initial input to the circuit, nor did we investigate the developmental origins

of central brain neurons, which receive the output from the circuit. We do note there is some evidence to support the generality that for assembly of feed-forward circuit motifs, presynaptic interneurons are born after their postsynaptic partners. Specifically, motor neurons are always circuit outputs (to muscle), and, in general, they are among the first neurons to be born during neurogenesis (*Meng and Heckscher, 2021*). Moreover, this pattern holds true in both *Drosophila* nerve cord and spinal cord (*Schmid et al., 1999*; *Fetcho and McLean, 2010*). Therefore, our data raise the possibility that one of many fundamental rules for circuit assembly is that feed-forward circuits are assembled sequentially from circuit output to circuit input.

## Materials and methods
### *Drosophila* strains and culture
Fly stocks were maintained at 25°C on standard cornmeal molasses medium. Experimental crosses were set up at the indicated temperatures. See below for details. See *Supplementary file 4* for details of fly lines.

### Immunohistochemistry
Larvae at different stages were dissected in Baines' solution. The dissected larval brains were adhered to poly-lysine-coated coverslip and fixed for 7 min in 4% paraformaldehyde solution (Electron Microscopy Services, Hatfield, PA) as previously described (*Heckscher et al., 2014*). Larval brains were then washed three times with phosphate-buffered saline containing 0.1% Triton X-100 (PBT), blocked 1 hr in PBT containing 2% normal goat serum (NBT), and stained with primary antibody in PBT at 4°C overnight. After washing, samples were stained with secondary antibody at room temperature for 1 hr, washed again, and applied with increasing percentages of ethanol in water (30, 50, 70, 95, 100) series to replace PBT. Samples were then immersed into xylene for clearance, and then mounted in DPX (Sigma, MI). Images were acquired using either a Zeiss LSM 800 confocal microscope, and were processed and analyzed using ImageJ. For a list of primary antisera, see *Supplementary file 4*. Secondary antibodies were obtained from ImmunoResearch (Bar Harbor, ME) and were used at a 1:400 dilution.

### Generation of the *pPL[KD]* transgene (Figure 1D)
The KDR-based permanent labeling plasmid, *pPL[KD]* (*nSyb* promoter and 5′UTR-KDRT-stop-KDRT-IVS-Syn21-GAL4-VP16-p10 3′UTR), was constructed in the backbone of nSybKOG (gift from Dr. Tzumin Lee) by restriction enzyme-mediated molecular cloning (*Awasaki et al., 2014*). The GAL4 coding region in nSybKOG was replaced by IVS-Syn21-GAL4-VP16-p10 3′UTR sequence. The IVS and p10 fragments were derived from the vector pJFRC81 (Addgene #36432), Syn21 sequence (*Pfeiffer et al., 2010*) was added by PCR amplification, and GAL4-VP16 was derived from the vector pBPGAL4.2::VP16Uw (Addgene #26228). The resulting *pPL[KD]* plasmid was incorporated into the docking site VK27 by integrase-mediated site-specific integration, performed by GenetiVision Corporation (Houston, TX). Details of molecular cloning and construct sequence are available upon request.

### Generation of the EMS-zp-AD transgene (Figure 1—figure supplement 1)
The pEntr-EMS plasmid was generated from a HiFi reaction (NEB) using the pEntr3c Gateway cloning (Invitrogen) plasmid digested with KpnI and NotI restriction endonucleases and four PCR reactions spanning the EMS promoter sequence found in *Estacio-Gómez et al., 2013*. The primer pairs used were (1) CGACTGGATCCGGTACCccagacagaactccatactccac and gtcgttaaacAAATGAATTGCCATAAGCG; (2) caattcatttGTTTAACGACCAACGCTC and tccggatggtCGAGCGGGATTTATGAGC; (3) atcccgctcgACCATCCGGATCTGGGCAAAAC and aatgaaaaccGTAAAAAATGCAGCCAACAAAGGG; (4) catttttttacGGTTTTCATTCCTTTTTGCG and GTCTAGATATCTCGAGTGCGGCCGCgtgtagtatggccgtcttctttgc.

The pEntr-EMS was combined with the Gateway cloning destination vector pBPzpGal4AD using an LR reaction to generate pBP-EMSzpAD.

### ts-MARCM experiments
In this study, we used ts-MARCM to determine both neuronal birth order and estimate neuronal birth time. *Birth order* is determined in reference to other neurons within a given lineage, for example,

first-born, second-born. Birth order is also often referred to as a neuron's temporal identity. Neuronal temporal identity is assessed by expression of temporal transcription factors. Consequently, determining birth order can be used to infer temporal transcription factor expression. *Birth time* refers to when during neurogenesis a neuron is born. Birth timing is usually reported in units of minutes or embryonic stage. Determining neuronal birth timing provides context into which a neuron is born (e.g., availability of other early-born neurons, transient signaling cues). In *Drosophila* embryos, neuron birth timing and birth order are not identical because during embryogenesis neuroblasts start to divide at different times. For example, NB3-5 is generated relatively early in embryogenesis and generates a large number of neurons, whereas MNB is generated much later and generates fewer neurons. And so, MNB can be generating first-born neurons at the same time that NB3-5 is generating much later-born neurons. We tried to birth date neurons using photo-activation and EdU/BrdU labeling, but were unsuccessful. So, instead, we estimated Basin birth timing using a series of carefully timed heat shocks to generate ts-MARCM clones. See below for details.

## EL-GAL4

We used the pan-EL driver, *EL-GAL4*, which is expressed in all ELs (*Heckscher et al., 2015*). Embryos of the genotype *hsFLP; UAS-rCD2.RFP, UAS-GFPi, FRT40A/UAS-mCD8-GFP, UAS-rCD2i, FRT40A; EL-GAL4/pPL[KD]* were used. Addition of the pPL[KD] amplifies expression from *EL-GAL4* (*Figure 1E* and *Figure 2B, C*). This optimization step was required to drive high levels of RNAi constructs that are at the core of the ts-MARCM strategy (e.g., UAS-rCDi). High RNAi levels are needed such that 'leaky' expression of membrane reporters (e.g., UAS-rCD3.RFP) is suppressed. Also, high levels of membrane-tethered reporter protein are needed to observe the detailed morphology of the entire neuron arbor.

### Birth order (*Figure 2*)

To determine the birth order of A1 ELs, embryos were collected for 2–4 hr intervals on apple juice plates. Collections were aged for 5–13 hr after egg collections. The collected samples were exposed to 37°C heat shock for 20–25 min. Heat shock stochastically induces FLP expression, which ultimately generates ts-MARCM clones. Heat-shocked samples were incubated at 29°C to boost GAL4 activity. Larvae were dissected at late L1 stage. Individual AI EL clones were visualized in samples with extremely sparse labeling (usually 1–2 clones per CNS) to obtain a clear morphology. For each EL clone, the generic EL identity of every labeled neuron was confirmed by Eve protein staining. The specific identity of each singly labeled EL neuron was determined by matching the morphology to connectome data (see below). Singly labeled neurons are the sole neuron expressing one fluorophore (either GFP or RFP). The remaining neurons expressed the opposite fluorophore, and we call these alternatively labeled neurons. To determine birth order of singly labeled ELs, we counted the number of alternatively labeled Eve(+) neurons. As a confirmation of assigned birth order, we also counted the number of unlabeled ELs (i.e., GFP[-], RFP[-], Eve[+]). To order A08e3, A08e2, and A083e1, we use the following logic. In our dataset, A08e2 was always singly labeled, and so A08e2 is mostly likely that last-born EL. Further, *Figure 1C* (and other examples) clearly shows both that A08e3 is the third from last-born and that later-born neurons are local. Therefore, we conclude A08e1 must be the second from last-born. The ts-MARCM clonal experiments were performed repeatedly until we assigned the birth order of all the A1 ELs. A total of 58 CNS were scored.

### Birth time (*Figure 12*)

The logic behind using ts-MARCM to determine birth timing is as follows: If a heat shock treatment occurred in embryos at the developmental stage either prior to during the cell division that generated a particular neuron type, then a clone will be induced. On the other hand, if the heat shock treatment occurred in embryos at the developmental stage after a neuroblast divides to give rise to a particular neuron, no clones will be generated. Therefore, determining birth timing requires more accurate staging of the developing embryos compared to experiments used to determine birth order.

Notably, in our ts-MARCM experiments there is a delay between when the heat shock is delivered to the embryo and the time when FLP recombinase protein has been produced and is active in dividing cells. To estimate this delay, we used ELs to 'calibrate' our assay. Specifically, we crossed ts-MARCM transgenes to *EL-GAL4*. We let flies lay eggs for 1 hr, then aged the egg collection for

7 or 9 hr before delivering a heat shock. In embryos aged 7 hr, the most frequently produced single-labeled EL is A08m. A08m is the fifth-born EL. The fifth-born EL is formed at ~9 hr of development (*Tsuji et al., 2008*; *Gunnar et al., 2016*; *Demilly et al., 2011*). In embryos aged 9 hr, the most frequently produced single-labeled EL was A08s. A08s is the eighth-born EL. The eighth-born EL is formed at ~11 hr of development (*Gunnar et al., 2016*; *Tsuji et al., 2008*). From these data, we infer that in our assay there is an ~2 hr delay between the time of heat shock and the time of FLP-induced recombination in a dividing cell.

## *BASIN-GAL4* (Figure 8)

To determine the birth order and birth timing of Basins, we used the pan-Basin driver, *72F11-GAL4*. Embryos of genotype *hsFLP; UAS-rCD2.RFP, UAS-GFPi, FRT40A/UAS-mCD8-GFP, UAS-rCD2i, FRT40A; 72F11-GAL4/+* were collected for 1 hr on apple juice plates. Collections were aged for 6–13 hr after egg collections. The collected samples were exposed to 37°C heat shock for 20, 30, or 35 min. Heat-shocked samples were then incubated at 29°C to boost GAL4 activity. Larvae were dissected at late L1 stage. We scored a total of 113 clones.

### Basins are lineage-related

We generated low levels of recombination at early times in development by providing 20 min heat shock at 6 or 7 hr after collection. A total of 12 CNSs were scored, and of those, 7 had one or two clones labeled. Of these seven CNSs, at least one of the two clones was a four-cell Basin clone.

### Birth order

Individual Basin clones were visualized in samples with sparse labeling in order to obtain a clear morphology. For each Basin clone, we identified the singly labeled neuron by matching it to connectome data. We birth ordered the singly labeled Basin by counting the number of alternatively labeled neurons in the clone.

### Birth timing

We determined when Basins were born. Basins are progeny of NB3-5. NB3-5 divides from stages 9–16 (*Schmid et al., 1999*). Each NB3-5 division gives rise to a GMC, which divides once to produce two progeny (*Monedero Cobeta et al., 2017*; *Moris-Sanz et al., 2014*). For embryos of each age, we scored the number of hemisegments displaying the following clonal morphologies:

- 0:4 clones. In 0:4 clones, all Basins have the same label, indicating the heat shock occurred before NB3-5 divided to generate the first Basin. 0:4 was the most common clone type for heat shocks applied at 6 or 7 hr. This corresponds to stage 12 embryos.
- 1:3, 1:2, and 1:1 clones. In 1:3 clones, three Basins are labeled with one marker, and the other Basin is singly labeled with the other marker. This indicates the heat shock occurred as NB3-5 was dividing to make the first Basin. In 1:2 clones, NB35 was dividing to make the second Basin. In 1:1 clones, NB3-5 was dividing to make third Basin. Clones of the 1:3, 1:2, and 1:1 types were most frequently observed in when heat shock was performed at 8, 9, and 10 hr. This corresponds to stages 13–15.
- 0:1 clones. In 0:1 clones, heat shock occurred either as NB3-5 was dividing to make the final Basin or as a GMC was dividing to make a Basin. In embryos heat-shocked at 11 hr, we find only 0:1 clones, suggesting at this time point only GMCs are dividing to produce Basins. This corresponds to late stage 15. We note that there are examples of 0:1 clones in all experiments that provided heat shocks from 6 to 11 hr. This could indicate some stochastic labeling. However, we also note that in embryos heat-shocked at ages 12 and 13 hr, Basin clones were never observed. Lack of clones indicates no cells were dividing to generate Basins.

Thus, peak Basin production occurs in embryos that have been heat-shocked after they were aged 8–10 hr. Taken together with the idea that there is an estimated delay of 2 hr from time of heat shock to FLP-induced recombination (see 'EL-GAL4' section above), this suggests that, in general, the majority of Basin interneurons are generated from NB3-5 between 10 and 12 hr of development, or late stage 13 to stage 15. Furthermore, these data suggest that peak Basin production is likely to occur within a 2-hr time window. This means in 2 hr NB3-5 produces four Basins, or one Basin every 30 min. In *Drosophila* larvae, neuroblasts are thought to divide with a frequency of once every 45 min.

Although we cannot definitively rule out the idea that production of one or more neuron type could sneak in, these data are consistent with the idea that Basins are continuously born.

To gain confidence that our heat shock experiment timing was robust across experiments and genotypes, we performed a similar set of experiments with *Basin-1-GAL4* (aka, *78F07-GAL4*) (*Figure 12*). *Basin-1-GAL4* labels only one Basin, which is distinct from *Basin-GAL4*, which labels all four Basins. Our *Basin-1-GAL4* and *Basin-GAL4* datasets are in general agreement, suggesting that our assay is robust across different genotypes and experiments. See below for details.

### *BASIN-1-GAL4* and *LADDER-D-GAL4* (Figure 12)

We used the neuron-specific drivers, *78F07-GAL4* and *20B01-GAL4*. These drivers expressed solely in Ladder D and Basin 1, respectively, but not in other neurons within the same lineage. Embryos of genotype *hsFLP; UAS-rCD2.RFP, UAS-GFPi, FRT40A/UAS-mCD8-GFP, UAS-rCD2i, FRT40A; 78F07-GAL4* and *hsFLP; UAS-rCD2.RFP, UAS-GFPi, FRT40A/UAS-mCD8-GFP, UAS-rCD2i, FRT40A; 20B01-GAL4* were used.

#### Birth timing

To determine the birth timing of Ladder D and Basin 1, embryos were collected within an hour interval on apple juice plates. They were aged for 6–13 hr after egg collection. The collected samples were exposed to 37°C heat shock for 30 min, and then incubated at 29°C to boost GAL4 activity. Larvae were dissected between late second larval instar and early third larval instar, within which the expression of the GAL4 drivers reach the peak of their intensities. Both Ladder D and Basin 1 can be clearly identified by the morphology (*Figure 12D and E*). The segmental identity was distinguished by Eve protein staining. We scored neuronal clones in segments A1–A7 based on the assumption that individual neurons should develop roughly at the same time in different abdominal segments, that is, Basin 1 in A1 and in A7 should form roughly simultaneously. This broader scope of scoring, together with the stronger clonal induction, maximized our capability to capture any larval CNS that had at least one stem cell or GMC undergoing division out of seven abdominal segments (or 14 hemi-segments). The last time point that we could catch larvae CNS with a ts-MARCM clone was last time point at which there could be a division to generate Ladder D or Basin 1. A total of 57 CNSs were scored.

### *BASIN-GAL4* and *EL-GAL4* (Figure 12)

We used both the pan-Basin driver, *72F11-GAL4*, and the pan-EL driver, *EL-GAL4*. Embryos of genotype *hsFLP; UAS-rCD2.RFP, UAS-GFPi, FRT40A/UAS-mCD8-GFP, UAS-rCD2i, FRT40A; 72F11-GAL4/EL-GAL4* were used.

#### Birth timing

To determine the relative birth order of Basins and ELs, embryos were collected for 1 hr on apple juice plates. Collections were aged for 7 or 9 hr after egg collections. The collected samples were exposed to 37°C heat shock for 33 min. The analysis was identical to that described for *Basin-GAL4* above. 29 CNSs were scored.

### Multi-color flip out (Figures 11 and 14)

We labeled single neurons using MCFO (*Nern et al., 2015*). MCFO stochastically labels with epitopes the membranes of cells within a GAL4 pattern. To obtain single-cell clones, adult flies laid for 24 hr on apple juice caps. Caps were heat-shocked in a water bath at 37–39°C for 15–30 min and incubated at 25°C for 4–5 hr. First-instar larvae were dissected. Their brains were stained for HA, Flag, and V5 epitopes to visualize single-cell clones. Larvae were also stained for Eve protein to confirm the identity of each single-cell clone as an EL interneuron or to assign segmental identity to each clone. We generated >100 single-cell clones and saw each neuronal morphology in a minimum of two separate larvae. Each clone was analyzed in dorsal and posterior views.

### Identification of stem cell lineage of Basins and Ladder D (Figures 9B–D and 10C–E)

To assign neurons to a specific neuroblast lineage, larvae of genotype *Basin-1-LexA/UAS-mCherry.NLS; NB3-5-GAL4/LexAop2-CsChrimson-mVenus* were used, where *NB3-5-GAL4* was one of three

different GAL4 lines (*49C03-GAL4, ham-GAL4, 59E09-GAL4*). We used three different GAL4s to label NB3-5 because any one GAL4 line might have unexpected off-target labeling. We reasoned that each line should have different off-target labeling, and so if we saw co-localization of markers in all three cases, this could be taken as strong evidence that Basin 1 was in the NB3-5 lineage. Lines also included FLP-based permeant labeling (UAS-FLP, *actin* promoter and FRT-stop-FRT-GAL4), which labels all progeny from a given neuroblast. Larvae were raised at 29°C on the apple juice plate. The larvae were then dissected in the L3 stage for further staining and imaging processing.

We used a similar logic to determine neuroblast origin for Ladders. Larvae of genotype *Ladder-D-LexA/UAS-mCherry.NLS; MNB-GAL4/LexAop2-CsChrimson-mVenus* were used, where *MNB-GAL4* was one of three different GAL4 lines (*13G03-GAL4, 70D06-GAL4*, and *65H11-GAL4*).

### Calcium imaging (Figure 7E–G)

For calcium imaging experiments, all larvae were within 6 hr of age on the day of recording and collected 48–54 hr after hatching. Larvae expressing GCaMP6m were rinsed with water and placed ventral side up on agarose pads with a 22 mm × 22 mm coverslip placed on top. Pads were made by pouring 3% agarose into a well. Recordings began with a 30 s period of no stimulus followed by a 30 s period of sound stimulus and ending with a final 30 s period of no stimulus. A Visaton FR12, 4 Ohm speaker (5 inches diameter) and a PYLE PCA2 stereo power amplifier was used to project sound. For further details, refer to *Marshall and Heckscher, 2022*. Images were acquired on a Zeiss LSM 800 confocal microscope using 0.1–0.2% 488 nm laser power with the pinhole entirely open. Images were acquired at 3 frames per second using a 10× (0.3 NA) or 20× (0.8 NA) objective. The calcium signal was continuously collected before, during, and after the stimulus. Extracting changes in GCaMP6m fluorescence amplitude was done using Fiji as in *Marshall and Heckscher, 2022*. A region of interest (ROI) that included the larval nerve cord was manually drawn, and the mean fluorescence within the ROI was calculated for each time point.

### Connectome analyses

The connectome dataset used in this study is a CNS reconstruction from a 6-hr-old first-instar larva, described in *Ohyama et al., 2015*.

### Identifying NB3-3A1L/R neurons in the connectome (*Figures 1–3*, *Supplementary file 1*)

In the connectome, the following ELs had been previously identified (A08x, A08m, A08c, A08s, A08e1, A08e2, A08e3) (*Wreden et al., 2017*; *Heckscher et al., 2015*). To identify other neurons in the lineage, we used the following logic: NB3-3A lineage has 13 neurons, including 2 non-EL neurons, 1 of which is a motor neuron and 11 EL interneurons (*Tsuji et al., 2008*; *Schmid et al., 1999*). We found that all EL interneurons form a bundle (*Figure 3*), and that the two non-EL neurons are also part of that bundle (*Figure 3A*). Therefore, in the connectome, we looked for a bundle that contained 13 neurons and that included all previously identified EL interneurons. This bundle is shown in *Figure 3C*. It contains two non-EL neurons—a motor neuron, an undifferentiated neuron, all previously identified ELs, and several interneurons that were candidates to be ELs. We confirmed these candidate neurons were ELs by making single-neuron clones with *EL-GAL4* driving ts-MARCM constructs and co-staining clones with anti-Eve (*Figure 2*). For details on matching neurons in light-level and connectome-level datasets, see *Wreden et al., 2017*; *Heckscher et al., 2015*. This confirmed the identity of A08j1, A08j3, A08j2, and A08o as ELs. While this article was in preparation, Mark et al. partially characterized multiple lineages, including NB3-3 (*Mark et al., 2021*). In agreement with our assignments, Mark et al. suggest that A08j1-A08j3 and A08o are ELs. Notably, our naming scheme for ELs differs slightly from that used by Mark et al. In our naming scheme, we use the first published name for each EL. Details can be found in *Supplementary file 5*.

### Identifying NB3-5A1 neurons in the connectome (*Figure 9*, *Supplementary file 3*)

In the connectome, Basins 1–4 in segment A1 had been previously identified (*Ohyama et al., 2015*; *Jovanic et al., 2016*). To identify other neurons in the lineage, we used the following logic: First, we found all neurons that bundled with Basin neurons (total of 34). Of these, 10 neurons had a medial

trajectory upon entering the neuropil. We excluded these neurons from further analysis for two reasons. (1) They are of a number and morphology that is consistent with their being from NB2-4 (*Schmid et al., 1999*). (2) The NB3-5 lineage is reported to have the capability of generating as many as 36 neurons (*Monedero Cobeta et al., 2017*). But, by late embryonic stage 17, only 19–24 are still found in abdominal segments (*Schmidt et al., 1997*). Excluding the 10 putative NB2-4 neurons from the lineage bundle left 24 neurons, which matched the number reported by Schmidt. Of these neurons, five were undifferentiated—with neurites ending prematurely with no synapse input or output. We excluded these from further analysis. The remaining 19 neurons fell into to broad morphological categories—neurons with dorsally projecting neurites and neurons with ventrally projecting neurites. Recently, for many lineages in the *Drosophila* nerve cord, dorsally projecting neurons were shown to belong to a Notch ON hemilineage and ventrally projecting neurons to a Notch OFF hemilineage (*Mark et al., 2021*). NB3-5 produces two hemilineages (*Monedero Cobeta et al., 2017*; *Moris-Sanz et al., 2014*). Of the remaining neurons in the lineage bundle, six 'Drunken' neurons had neurite trajectories that diverge dorsally after entering the neuropil. We consider these to be potentially Notch ON neurons from the NB3-5 lineage and excluded them from our analysis. We were left with 13 neurons in the bundle, which also contained Basins. All of these neurons had ventral trajectory in the neuropil (*Figure 9E*). These 13 neurons were used for analysis (*Figure 9F–H*, *Supplementary file 3*).

## Identifying Ladder neurons and NB7-1 interneurons in the connectome (*Figures 10 and 14*, *Supplementary file 2*)

In the connectome, Ladders A–F in segment A1 had been previously identified (*Jovanic et al., 2016*). See *Supplementary file 2* for Ladder inputs. NB7-1 interneurons had already been identified (*Mark et al., 2021*).

## Neuronal reconstruction, finding synaptic partners, and proof reading annotations

Once all the NB-3-3A1 neurons were identified, their skeletons were reviewed to greater than 90% with mainly areas in the brain unreviewed. Every skeleton with an input synapse onto a NB3-3A1L/R neuron was reconstructed in an attempt to generate complete neurons with cell bodies. Upstream left/right neuron pairs were identified by mirror image morphology and synapse similarity. Upstream pairs that had input onto NB3-3A1L/R pairs at a greater than four or more synapses on one of the pair and two or more synapses on the other neuron of the pair were reviewed to greater than 80% (*Supplementary file 1*).

## Calculating neuron birth order versus proximal neurite length (*Figures 3E, 9H and 14D*)

The distance was calculated using the CATMAID function 'Measure the distance between two nodes.' The two nodes used were the cell body and a node chosen by eye where the skeleton entered the neuropil.

## Calculating inputs onto NB3-3A1L/R by type (*Figure 4A*, *Supplementary file 1*)

Input neurons onto NB3-3A1L/R neurons were broadly categorized into four types. Nerve cord interneurons had a cell body in segments T1-A10. Sensory neurons had no cell body, but instead had axons entering the nerve cord in bundles from the periphery. They also show a unique, electron-dense cytoplasm. Central brain and SEZ neurons had cell bodies anterior to the nerve cord. Unknown included fragments of neurons that could not be traced back to cell bodies.

## Calculating sensory neuron inputs onto NB3-3A1L/R (*Figure 4B*)

A sensory neuron was determined to be chordotonal, proprioceptive, or other, based on morphology of the central axon (*Grueber et al., 2007*; *Heckscher et al., 2015*; *Ohyama et al., 2015*).

## Generating histograms of distribution of synapses (*Figures 4C and 6F*)

All interneurons or sensory neurons that made synapses onto a NB3-3A1L or NB3-3A1R were included in the analysis (760 neurons, *Supplementary file 1*) in *Figure 4C*. Histogram bins were 1, 2, 3, 4, 5, 6, 7, 8, 9, or 10+ synapses. In *Figure 6F*, all neurons that made synapses onto any NB3-3A1L or NB3-3A1R were included in analysis (*Supplementary file 2*). Histogram bins were 0–4, 5–9, 10–14, 15–19, 20–24, 25–29, 30–34, 35–39, 40–44, and 45+.

## Calculating L/R paired interneuron inputs onto NB3-3A1L/R (*Figure 4*)

We display only left–right 'L/R' paired interneurons. We consider L/R paired interneurons to include the subset of nerve cord interneurons that matched the following criteria. (1) For hemisegmental homologs (i.e., left–right pairs of neurons), one neuron formed >3 synapses onto a NB3-3A1L or NB3-3A1R neuron and the other neuron formed >1 synapse onto the hemisegmentally homologous NB3-3A1L or NB3-3A1R neuron. (2) For unpaired midline neurons, the neuron formed >3 synapses with a NB3-3A1L or NB3-3A1R neuron and formed >1 synapse with the hemisegmentally homologous NB3-3A1L or NB3-3A1R neuron. These neurons are listed as L/R paired interneurons in *Supplementary file 1*. In total, this was a total of 198 nerve cord interneurons.

## Calculating synapses from nerve cord neurons onto early-born ELs (*Figure 6E*, *Supplementary file 2*)

Starting with all inputs to NB3-3A1L/A1R, we found the subset of inputs onto early-born ELs (548 neurons, 2618 synapses). Of those, we identified interneurons and sensory neurons (see calculating inputs onto NB3-3A1L/R by type) (334 neurons, 2061 synapses). Of these, neurons were binned into classes (Chordotonals, Ladders, Basins, early-born ELs) based on previously described morphological criteria (*Ohyama et al., 2015*; *Jovanic et al., 2016*; *Wreden et al., 2017*).

## Analysis of highly connected neurons (*Figure 13*)

Highly connected neurons were defined as follows: (1) Sensory neuron or interneuron of the nerve cord, (2) left–right paired or unpaired, if midline neuron (e.g., Ladder), and (3) one member of the pair made 10 or more synapses onto early-born ELs in segment A1. 10 or more synapse level was chosen because the 62 neurons contributing 10 or more synapses onto early-born ELs account for a majority (58% or 1114/1935) of all synapses from sensory or interneurons that are made onto early-born ELs in A1. The lineage origin of a neuron of highly connected neurons in the A1 segment was deduced by first identifying the bundle of neurons with which it fasciculates as it enters the neuropil. Additional information on lineage identification relied on the number of neurons in the fascicle, the projection of those neurons in the neuropil, and the position of the fascicle within the hemisegment. This information was then compared to previous studies that examined neuron lineage assignment (*Bossing et al., 1996*; *Schmidt et al., 1997*; *Schmid et al., 1999*; *Mark et al., 2021*). For neurons outside of A1, where connectome annotations are not as complete as in A1, a similar methodology was used except that the shape of the neuron and the shape, size, and position of the neuron bundle were compared to lineages in A1 to identify the lineage.

### Distance analyses

## Nonbinary analysis (*Figures 5A, C–E, 9F and G and 10F*)

We treated inputs to a neuron as a vector, which contained the number of synaptic contacts. The order of inputs was the same for every neuron. We calculated the Euclidean distance between the two inputs vectors. These distance vectors summarize how similar the two neurons' inputs are. To test for significance, we created 100 surrogate vectors. We permuted the inputs of each neuron independently, preserving the number of inputs (in degree) but shuffling their identity. We then calculated the Euclidean distances of pairs in the shuffled vectors, building a distribution of Euclidean distances expected by chance. We normalized real data to shuffled data as follows: real-(mean[shuffled])/(standard deviation [shuffled]), which generated z-scores. Z-scores of >+1.96 or <–1.96 were considered significantly more similar or different, respectively, than would be observed by chance (at the alpha = 0.05 level). Significant pairwise differences and similarities were defined as significant positive and negative (respectively) z-scores of the real Euclidean distance, at p<0.05 (two-tailed).

## Binary analysis (Figure 5)

Since we observed a broad variation in synaptic strengths and input onto ELs, to disambiguate the identity of inputs from their respective strength, we repeated distance analyses using input identity only. To do so, we treated inputs to a neuron as a binary vector, with 1 for the presence and 0 for the absence of input, respectively. Otherwise, analysis was identical to nonbinary. Broadly speaking, results from binary and nonbinary analyses were similar. One notable exception is that binary analysis did not identify statistically significant differences.

### Left–right and following pair analysis (*Figure 5B*)

We examined the normalized (z) Euclidean distance scores obtained from binary connectomes regardless of their significance in two types of neuron pairs separately: Left–right pairs are comprised of the same neurons in the left and right hemisegment, for example, A08j2 A1L and A08j2 A1R. The following pairs are comprised of neurons that are born consecutively in the same hemisegment, for example, A08j3 A1L and A08j2 A1L. In *Figure 5B*, we pooled the scores of the two pair types over early-born, late-born, and undifferentiated and motor neurons separately.

## Acknowledgements

We thank the Janelia Fly EM Project Team for the gift of the EM volume, and the Janelia visitor program for their support. We would like to thank the following for contributing to EM annotation: Akria Fushiki, Albert Cardona, Andreas Schoofs, Antia Burgos, Aref Arzan Zarin, Avinash Khandelwal, Brittany Kemp, Casey M Schneider-Mizell, Chris Q Doe, Elizabeth Barsotti, Ingrid Andrade, Jamie Macleod, Larisa Maier, Laura Herren, Maarten F Zwart, and Xinyu Tang. This work was supported by NIH grants NS105748 (to ESH), EY022338 (to JNM), and T32 HD044164 (to ZDM), and NSF grant DGE-1746045 (to JLM).

## Additional information

### Funding

| Funder | Grant reference number | Author |
| --- | --- | --- |
| National Institute of Neurological Disorders and Stroke | NS105748 | Ellie Heckscher |
| National Eye Institute | EY022338 | Jason MacLean |
| Eunice Kennedy Shriver National Institute of Child Health and Human Development | T32 HD044164 | Zarion D Marshall |
| National Science Foundation | DGE-1746045 | Julia L Meng |

The funders had no role in study design, data collection and interpretation, or the decision to submit the work for publication.

### Author contributions

Yi-wen Wang, Chris C Wreden, Investigation, Writing – original draft; Maayan Levy, Investigation, Methodology; Julia L Meng, Zarion D Marshall, Investigation; Jason MacLean, Methodology, Writing – review and editing; Ellie Heckscher, Conceptualization, Funding acquisition, Supervision, Writing – original draft, Writing – review and editing

### Author ORCIDs

Chris C Wreden ⬚ http://orcid.org/0000-0002-3083-8790
Ellie Heckscher ⬚ http://orcid.org/0000-0001-7618-0616

Decision letter and Author response
Decision letter https://doi.org/10.7554/eLife.79276.sa1
Author response https://doi.org/10.7554/eLife.79276.sa2

## Additional files

### Supplementary files

• Supplementary file 1. Inputs onto NB3-3A1L/R neurons. With following tabs: All inputs on NB3-3A1L/R neurons (*Figure 4A*). Left–right paired interneuron inputs (*Figure 4D*). NB3-3 to NB3-3 connectivity (*Figure 11G*).

• Supplementary file 2. Inputs onto early-born ELs. With following tabs: All sensory neuron and interneuron inputs onto early-born ELs (*Figure 6E*). Summary (*Figure 6EB*). Inputs onto early-born ELs (*Figure 6B*). Input onto each early-born EL (*Figure 7B–D*).

• Supplementary file 3. Inputs onto Basins and Ladders. With following tabs: Basin A1 connectivity (*Figures 9F and G and 11G*) Basin names (*Figure 9*). Ladder A1 connectivity (*Figures 10F and 11G*).

• Supplementary file 4. Resources. With the following tabs: Genotypes used in this study. Fly lines antibody list.

• Supplementary file 5. Names for ELs in A1.

• Supplementary file 6. Summary of other highly connected interneurons.

• Transparent reporting form

### Data availability

All data generated or analyzed during this study are included in the manuscript and supporting files (Supplementary files 1-6) Source data files are provided for Figures 3,4,5,6,7,9,10,13,14.

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
