## [Editor Report]

The article presents a thorough analysis of specific neuronal lineages in the early larval ventral nervous system that relates the birth order to circuit connectivity and function. The key findings of the work are (1) the identification of sharp temporal cohort divisions for the lineages under investigation, (2) synapse formation between neurons of different lineages and temporal cohorts, and (3) the observation that output neurons in this instance are born prior to input neurons.

---

## [Decision Letter]

**Decision letter after peer review:**

Thank you for submitting your article "Sequential addition of neuronal stem cell temporal cohorts generates a feed-forward circuit in the *Drosophila* larval nerve cord" for consideration by *eLife*. Your article has been reviewed by 3 peer reviewers, including P Robin Hiesinger as Reviewing Editor and Reviewer #1, and the evaluation has been overseen by Ronald Calabrese as the Senior Editor.

Essential revisions:

All three reviews agreed on the high quality of the data, but also raised very similar concerns, leading to clear suggestions for a revision as outlined below.

The reviewers' shared concerns regard (1) the generalized statements on the role of birth order for connectivity, and (2) the generalized statements about output neurons being born first.

Regarding concern (1) reviewer #2 suggested: 'To show that birth-time is important for the assembly of the feedforward circuit, can the authors manipulate the relative birth time of the output vs input neurons (e.g. using mis-expression of TTFs) and see whether the feedforward circuit can still be assembled?'

Regarding concern (2) reviewer #3 suggested '~34% of the interneuron inputs to the EL-early neurons remain unmapped. This includes six neurons that synapse onto the EL-early neurons over 10 times each. It is therefore likely that other lineages contribute neurons that synapse onto the EL-early neurons. […] Lineage-mapping and birth-dating the six neurons with significant synaptic input onto the EL-early neurons would be very helpful as this data would either validate or challenge the conclusions that (1) output neurons are born first and (2) only three lineages contribute to this circuit. As a first step, one could determine whether the cell bodies of the six unidentified neurons are clustered together (which would suggest that they are generated by the same neuroblast). '

On balance, the reviewers felt that the experimental suggestion regarding concern 1 above would very substantially strengthen the paper, but it is not quaranteed that this experimental approach would be feasible. The authors are thus encouraged to explore this possibility, but a success of such an experiment is not deemed essential.

Regarding concern (2), the reviewers agreed that an experimental attempt should be made to determine lineage-mapping and birth-dating for at least one or some of the other significant input neurons to support the core conclusion of the paper.

*Reviewer #1 (Recommendations for the authors):*

On balance, the quality and relevance of the findings clearly outbalance the weaknesses, which I think can mostly, or completely, be addressed with textual changes. Specifically, I would like the authors to address the following points:

(1) Regarding main conclusion (2), the data showing most synapses with three inputs (Basin, Ladder, A08) onto the chosen early-born Els appears clear, as does their lineage analysis. The interpretation of 'highly selective wiring among temporal cohorts' appears less clear to me. First, the wording 'among' may still be read as preferential connectivity within each of the temporal cohorts, so the intended meaning of preferential connectivity between different temporal cohorts could be written with more clarity.

Conceptually, the discovery that the three main partners are derived from different lineages, seems a little less surprising to me than presented. Specifically, the authors note that 'Further, we note that temporal cohorts that form synapses on early-born ELs come from just three of the 30 neuroblast in the nerve cord, demonstrating high selectivity.' seems an odd emphasis. The three main synaptic cell types have to each come from some neuroblast lineage. They come from three different ones. That there are 27 others does not argue for or against any specificity. My more simplistic interpretation would be that neurons that are from the same lineage and born around the same time are more similar to each other than other neurons (as shown by authors) simply because their transcriptional program is still more similar. Why this should lead to connectivity amongst themselves is, at least to me, even theoretically entirely unclear (and is indeed not the case). Hence, I would say the findings are rather expected than surprising – synaptic partnerships are between neurons from some not necessarily related lineages – and amongst 30 lineages, the likelihood of three being from three different ones is rather high…

To generalize this idea, a larger number of neurons would need to be investigate, which I consider beyond the scope of this study. So maybe the authors can just adjust the presentation, discussion, or educate me!

(2) Regarding main conclusion (3), i.e. output neurons are born first, I find the statement 'Further, this rules out the strict (early-early late-late) hypothesis that neurons born at the same time are circuit partners.' also too strong.

First, 1 of only 3 input neurons they looked at, the A08 interneurons, are also early-born Els and thus born at the same time – in accordance with the zebrafish example/hypothesis. This is really a low sampling rate – if 2 of the 3 were born at the same time the authors would have had to conclude that the majority behaved like the principle presented in the zebrafish data… Not sure how 'dogmatic' this zebrafish model really is – it is not exactly my field – but it seems to be just an example of a set of principles found in fish, not a dominant model in the field? It seems to me the authors present a new, well-documented instance where two out of three presynaptic neurons are later born than the downstream connected neurons. I really do not see what can be generalized from these numbers.

Another rather strongly worded conclusion is the notion of input/output. Yes, in this specific partnership under investigation Basin, Ladder and A08 are presynaptic to these early-born Els, but they are all interneurons with further inputs and outputs. Do the authors think that the next output level will again be born earlier? How general is this principle supposed to be? AS it is, this (interesting and well-documented) finding is way too far from (quote: ' a fundamental strategy for assembling feed-forward circuits'). I think the authors' study is important and should be published without the need for such claims. Again, happy to learn if I am missing something.

*Reviewer #2 (Recommendations for the authors):*

1) The authors can discuss what might underlie the sharp transition between temporal cohorts. What is the correlation between TTF (Hb, Kr, Pdm, Cas) expression with the border of the temporal cohorts, specifically between the transition from early EL or late ELs?

2) The authors showed that in a feed-forward circuit, early-born EL neurons which are the output neurons are born before input neurons (Ladder and Basin interneurons). They also mentioned in the discussion that motor neurons which are output neurons are among the first-born neurons in the ventral nerve cord and spinal cord. However, questions still remain whether output neurons are born before input neurons in feed-forward circuits is a general mechanism or not. What are the hypotheses about the importance of the birth-order of output vs input neurons? To show that birth-time is important for the assembly of the feedforward circuit, can the authors manipulate the relative birth time of the output vs input neurons (e.g. using mis-expression of TTFs) and see whether the feedforward circuit can still be assembled?

*Reviewer #3 (Recommendations for the authors):*

1) Figure 1: the ts-MARCM clones should be moved to the main results (from supplemental) since the birth order of neurons in the EL lineage is an important finding of the paper.

2) Lines 217-235, Figure 4: The network science-based approach is difficult to follow in the text and the figure. A clearer explanation of the methodology and results is needed to help the reader follow along. Figure 4B is particularly hard to understand.

3) The calcium imaging data (Figure 5 J-L) could be moved to supplemental (especially since similar data was shown in a previous paper, though with a less sensitive tool).

4) Lineage-mapping and birth-dating the six neurons with significant synaptic input onto the EL-early neurons would be very helpful as this data would either validate or challenge the conclusions that (1) output neurons are born first and (2) only three lineages contribute to this circuit. As a first step, one could determine whether the cell bodies of the six unidentified neurons are clustered together (which would suggest that they are generated by the same neuroblast).

5) Are the temporal cohorts of the EL-early, Basin and Ladder neurons that wire together generated from the same tTF window? This would be very interesting to investigate as it could provide a model for how neurons find each other in the developing circuit.

6) The definition of a feed-forward circuit should be included in the Results discussion when it is first reported (currently it is defined in the Discussion). This will help readers who may not be familiar with the term.

7) Transparent black boxes are visible in the panels of Fig6 C and D. This is likely a formatting error but should be corrected.

---

## [Author Response]

Essential revisions:All three reviews agreed on the high quality of the data, but also raised very similar concerns, leading to clear suggestions for a revision as outlined below.The reviewers' shared concerns regard (1) the generalized statements on the role of birth order for connectivity, and (2) the generalized statements about output neurons being born first.Regarding concern (1) reviewer #2 suggested: 'To show that birth-time is important for the assembly of the feedforward circuit, can the authors manipulate the relative birth time of the output vs input neurons (e.g. using mis-expression of TTFs) and see whether the feedforward circuit can still be assembled?'Regarding concern (2) reviewer #3 suggested '~34% of the interneuron inputs to the EL-early neurons remain unmapped. This includes six neurons that synapse onto the EL-early neurons over 10 times each. It is therefore likely that other lineages contribute neurons that synapse onto the EL-early neurons. […] Lineage-mapping and birth-dating the six neurons with significant synaptic input onto the EL-early neurons would be very helpful as this data would either validate or challenge the conclusions that (1) output neurons are born first and (2) only three lineages contribute to this circuit. As a first step, one could determine whether the cell bodies of the six unidentified neurons are clustered together (which would suggest that they are generated by the same neuroblast). 'On balance, the reviewers felt that the experimental suggestion regarding concern 1 above would very substantially strengthen the paper, but it is not quaranteed that this experimental approach would be feasible. The authors are thus encouraged to explore this possibility, but a success of such an experiment is not deemed essential.Regarding concern (2), the reviewers agreed that an experimental attempt should be made to determine lineage-mapping and birth-dating for at least one or some of the other significant input neurons to support the core conclusion of the paper.

Regarding concern 1, manipulating temporal transcription factors, we think this is a wonderful suggestion, but it is outside of the scope of this paper. As you likely know, we have already published two full *eLife* papers in which we manipulate temporal transcription factors and examine motor neuron to muscle connectivity patterns (Meng et al., 2019, 2020). In the context of interneuron to interneuron connectivity, as described in this paper, temporal transcription factor manipulations would be highly informative. However, on a technical level, these experiments would require complex stock building, which would take months. Additionally, exactly how we would assay any resulting changes in synaptic connectivity is a non-trivial problem, likely involving more connectomics work. Thus, experiments could take years to complete. Furthermore, from an intellectual perspective, when we think of temporal transcription factor manipulations, they help us understand how the system responds to specific types of perturbations, but not the naturally-occurring correlations between connectivity, lineage, birth order, and birth timing, which is the goal of this work. Further, as it stands now this paper is fourteen figures long. We do hope to do these experiment soon, and will report on those experiments.

Regarding concern 2, there were six neurons in question, each of which had substantial input onto early-born ELs. We used a connectomic approach to infer the lineage origin and birth timing within the lineage for each. We find that likely they each belong to a different lineage and most are later-born (Figure 13). One of the neurons, “A03 upstream of A08m” was inferred to be progeny of NB7-1. To confirm this inference, we made single neuron clones using an available NB7-1-GAL4 driver (Figure 14). Then, to birth date the neuron, we used a connectomics approach, measuring the cortex neurite length as a proxy for birth time within the lineage (Figure 14). We find that “A03 upstream of A08m” is born on or after the 12th division of NB7-1 (Figure 14). The early-born ELs are born from NB3-3 at the time when NB7-1 is performing the fifth to ninth division. We conclude that “A03 upstream of A08m” must be born after early-born ELs (Figure 14). Furthermore, we find that three other interneurons from the NB7-1 lineage also synapse onto early-born ELs, albeit weakly. And these NB7-1 interneurons must also be later born than early-born ELs. Unfortunately, there are not good reagents available for us to conduct similar experiments for the remaining five neurons, which come from other lineages.

Reviewer #1 (Recommendations for the authors):On balance, the quality and relevance of the findings clearly outbalance the weaknesses, which I think can mostly, or completely, be addressed with textual changes. Specifically, I would like the authors to address the following points:(1) Regarding main conclusion (2), the data showing most synapses with three inputs (Basin, Ladder, A08) onto the chosen early-born Els appears clear, as does their lineage analysis. The interpretation of 'highly selective wiring among temporal cohorts' appears less clear to me. First, the wording 'among' may still be read as preferential connectivity within each of the temporal cohorts, so the intended meaning of preferential connectivity between different temporal cohorts could be written with more clarity.

Thank you for pointing this out. We have updated the text throughout according to this suggestion.

Conceptually, the discovery that the three main partners are derived from different lineages, seems a little less surprising to me than presented. Specifically, the authors note that 'Further, we note that temporal cohorts that form synapses on early-born ELs come from just three of the 30 neuroblast in the nerve cord, demonstrating high selectivity.' seems an odd emphasis. The three main synaptic cell types have to each come from some neuroblast lineage. They come from three different ones. That there are 27 others does not argue for or against any specificity. My more simplistic interpretation would be that neurons that are from the same lineage and born around the same time are more similar to each other than other neurons (as shown by authors) simply because their transcriptional program is still more similar. Why this should lead to connectivity amongst themselves is, at least to me, even theoretically entirely unclear (and is indeed not the case). Hence, I would say the findings are rather expected than surprising – synaptic partnerships are between neurons from some not necessarily related lineages – and amongst 30 lineages, the likelihood of three being from three different ones is rather high…To generalize this idea, a larger number of neurons would need to be investigate, which I consider beyond the scope of this study. So maybe the authors can just adjust the presentation, discussion, or educate me!

The old text on lines 410-412 read: “Further, we note that temporal cohorts that form synapses on early-born ELs come from just three of the 30 neuroblast in the nerve cord, demonstrating high selectivity”. This has been replaced by the following text, on lines 449-455:

“Further, we note that temporal cohorts that form synapses on early-born ELs come from three of the 30 neuroblast in the nerve cord. This finding rules out the following two models: (1) that morphologically similar neurons from different lineages synapse onto early-born ELs, and (2) that multiple temporal cohorts from a single neuronal stem cell synapse onto early-born ELs. Instead, it shows that neurons from a limited number of neuronal stem cell lineages selectively synapse onto early-born ELs.”

(2) Regarding main conclusion (3), i.e. output neurons are born first, I find the statement 'Further, this rules out the strict (early-early late-late) hypothesis that neurons born at the same time are circuit partners.' also too strong.

Thank you for pointing this out. What we meant by using the word “strict” was that early-early and late-late was the only mechanism at play and that our data rule out such a possibility. The old text on lines 456-458 read: “More generally, this demonstrates that circuit outputs are born before circuit inputs. Further, this rules out the strict (early-to-early late-to-late) hypothesis that neurons born at the same time are circuit partners.” This has been replaced by the following text on lines 503-506:

“More generally, this demonstrates that circuit outputs from one lineage are born before circuit inputs from other lineages. Further, this rules out an exclusive early-to-early hypothesis that states that neurons born at the same time are circuit partners.”

We also provide a more nuanced discussion of the early-to-early late-to-late hypothesis on lines 459-471:

“One model for motor circuit development suggests that early-born neurons wire with each other to generate circuits driving fast movements, whereas later-born neurons wire with each other to generate circuits driving refined movement (i.e., early-to-early and late-to-late model) (Fetcho and McLean, 2010)(Ampatzis et al., 2014)(Mark et al., 2021). Furthermore, we found that early-born ELs from different segments synapse with each other, which raised the possibility that an early-to-early model could also explain the observed pattern of wiring. However, there is suggestive evidence that an exclusive early-to-early model cannot explain the full complexity of wiring patterns seen in motor circuits (Kishore et al., 2014)(Menelaou et al., 2014)(Song et al., 2020). Thus, here, we test the hypothesis that early-born ELs wire with neurons from other lineages, which born at the same time as early-born ELs.”

First, 1 of only 3 input neurons they looked at, the A08 interneurons, are also early-born Els and thus born at the same time – in accordance with the zebrafish example/hypothesis. This is really a low sampling rate – if 2 of the 3 were born at the same time the authors would have had to conclude that the majority behaved like the principle presented in the zebrafish data… Not sure how 'dogmatic' this zebrafish model really is – it is not exactly my field – but it seems to be just an example of a set of principles found in fish, not a dominant model in the field? It seems to me the authors present a new, well-documented instance where two out of three presynaptic neurons are later born than the downstream connected neurons. I really do not see what can be generalized from these numbers.

When thinking about sampling rate, consideration of units is important. For the unit of lineages, we had previously examined 3. But, for the unit of neurons, we examined over 300, and for the unit of synapses, we examined over 1900 (Figure 6E, Table S2). Nonetheless, even when focusing at the unit of lineages, we now include new data (Figures 13, 15), which pushes the observation to 5 of ~7 lineages (71%). We also include further context about the observation that most neurons that synapse onto early-born ELs are born after early-born ELs. In the Discussion, on lines 759-764:

“However, from a first principles perspective, we would expect the opposite. This is because 22 of 30 neuroblasts in the nerve cord are born before NB3-3, and many of them divide multiple times to produce neurons before NB3-3 begins to generate early-born neurons (Doe 1992). Therefore, there should be more neurons born before early-born ELs compared to those born after early-born ELs. And so, by chance alone we would expect early-born ELs to get more input from neurons born earlier or at the same time.”

Another rather strongly worded conclusion is the notion of input/output. Yes, in this specific partnership under investigation Basin, Ladder and A08 are presynaptic to these early-born Els, but they are all interneurons with further inputs and outputs. Do the authors think that the next output level will again be born earlier? How general is this principle supposed to be? AS it is, this (interesting and well-documented) finding is way too far from (quote: ' a fundamental strategy for assembling feed-forward circuits'). I think the authors' study is important and should be published without the need for such claims. Again, happy to learn if I am missing something.

Thank you for this comment. To address it we temper our language throughout the text, and we add section to the Discussion (lines 781-806):

“In what circumstances does sequential addition of temporal cohorts from different lineages occur? Studies of circuit assembly are still in their infancy. It is known that lineage-circuit relationships differ depending on circuit anatomy. But, the converse is not known. That is, do all circuits of the same anatomy have a common lineage-circuit relationship? In the case of this study, we focused on a single feed-forward circuit in the *Drosophila* larval nerve cord. And the specific question it raises is: To what extent is sequential addition of temporal cohorts from different lineages the only mechanism used to assembly all feed-forward motifs? Currently, our answers are only partial and speculative. We note that there are so many connections made among neurons, it is unlikely that just one simple phenomenon that can explain the full complexity. The goals of current research must be to identify rules and the circumstances where those rules apply. For example, our data rule out a “strict” early-to-early, late-to-late model, meaning that this model alone cannot explain the wiring we observe in this circuit. And yet, an early-to-early model does apply to inter-segmental wiring among temporal cohorts of the same lineage. Further, this early-to-early wiring phenomenon occurs alongside a sequential addition phenomenon. We speculate there could be additional phenomena underlying assembly of this simple motif. For example, beyond local interneurons, we did not determine the birth date of sensory neurons, which provide the initial input to the circuit, nor did we investigate the developmental origins of central brain neurons, which receive the output from the circuit. We do note there is some evidence to support the generality that for assembly of feed-forward circuit motifs, pre-synaptic interneurons are born after their post-synaptic partners. Specifically, motor neurons are always circuit outputs (to muscle) and in general they are among the first neurons to be born during neurogenesis (Meng and Heckscher, 2021). Moreover, this pattern holds true in both Drosophila nerve cord and spinal cord (Schmid et al., 1999)(Fetcho and McLean, 2010). Therefore, our data raise the possibility that one of many fundamental rules for circuit assembly is that feed-forward circuits are assembled sequentially from circuit output to circuit input.”

Thank you again for your time and thoughts. We hope we have addressed your concerns.

Reviewer #2 (Recommendations for the authors):Major points:1) The authors can discuss what might underlie the sharp transition between temporal cohorts. What is the correlation between TTF (Hb, Kr, Pdm, Cas) expression with the border of the temporal cohorts, specifically between the transition from early EL or late ELs?

We updated the Discussion to address this point (line 709-720):

“A third open question is what sets up the borders of temporal cohorts. Previous work used motor neuron temporal cohorts of NB7-1 and NB3-1 as a model to address this question (Meng et al., 2019)(Meng et al., 2020). In these stem cells, mis-expression of temporal transcription factors, Hunchback, Pdm, and Castor modulates the number of motor neurons in a temporal cohort, without changing the size of the lineage. Thus, temporal transcription factors are able to regulate motor neuron temporal cohort borders. But, it remains unclear how temporal transcription factors do so. Two, not mutually exclusive possibilities are that they act as transcriptional co-factors to induce differential gene expression programs, and/or that they act as pioneer factors to alter the chromatin landscape. For NB3-3, we note the during the seventh to eleventh divisions, which generates late-born ELs, NB3-3 expresses Grainyhead (Tsuji et al., 2008), which raises the possibility that Grainyhead may define the late-born EL cohort. Testing this idea is an important future direction.”

2) The authors showed that in a feed-forward circuit, early-born EL neurons which are the output neurons are born before input neurons (Ladder and Basin interneurons). They also mentioned in the discussion that motor neurons which are output neurons are among the first-born neurons in the ventral nerve cord and spinal cord. However, questions still remain whether output neurons are born before input neurons in feed-forward circuits is a general mechanism or not. What are the hypotheses about the importance of the birth-order of output vs input neurons? To show that birth-time is important for the assembly of the feedforward circuit, can the authors manipulate the relative birth time of the output vs input neurons (e.g. using mis-expression of TTFs) and see whether the feedforward circuit can still be assembled?

We address the question of whether the phenomenon observed in this study applies to other circuits in a new section of the Discussion (lines 781-806):

“In what circumstances does sequential addition of temporal cohorts from different lineages occur? Studies of circuit assembly are still in their infancy. It is known that lineage-circuit relationships differ depending on circuit anatomy. But, the converse is not known. That is, do all circuits of the same anatomy have a common lineage-circuit relationship? In the case of this study, we focused on a single feed-forward circuit in the *Drosophila* larval nerve cord. And the specific question it raises is: To what extent is sequential addition of temporal cohorts from different lineages the only mechanism used to assembly all feed-forward motifs? Currently, our answers are only partial and speculative. We note that there are so many connections made among neurons, it is unlikely that just one simple phenomenon that can explain the full complexity. The goals of current research must be to identify rules and the circumstances where those rules apply. For example, our data rule out a “strict” early-to-early, late-to-late model, meaning that this model alone cannot explain the wiring we observe in this circuit. And yet, an early-to-early model does apply to inter-segmental wiring among temporal cohorts of the same lineage. Further, this early-to-early wiring phenomenon occurs alongside a sequential addition phenomenon. We speculate there could be additional phenomena underlying assembly of this simple motif. For example, beyond local interneurons, we did not determine the birth date of sensory neurons, which provide the initial input to the circuit, nor did we investigate the developmental origins of central brain neurons, which receive the output from the circuit. We do note there is some evidence to support the generality that for assembly of feed-forward circuit motifs, pre-synaptic interneurons are born after their post-synaptic partners. Specifically, motor neurons are always circuit outputs (to muscle) and in general they are among the first neurons to be born during neurogenesis (Meng and Heckscher, 2021). Moreover, this pattern holds true in both Drosophila nerve cord and spinal cord (Schmid et al., 1999)(Fetcho and McLean, 2010). Therefore, our data raise the possibility that one of many fundamental rules for circuit assembly is that feed-forward circuits are assembled sequentially from circuit output to circuit input.”

We prefer to view the fact that we have raised the question of generality not as a weakness of the paper, but as one of the impacts of the paper.

We address the potential importance of birth order in the Discussion, lines 765-779:

“What are the hypotheses about the importance of birth-order of output versus input neurons? In *Drosophila*, birth order is linked to two things: (1) lineage intrinsic factors such as dynamically changing programs of gene expression in the stem cell, (2) lineage extrinsic factors or the dynamic environmental context into which neurons are born. Intrinsic factors, or extrinsic factors, or both may be playing a role in the assembly of this feed forward circuit. A potential intrinsic mechanism is that of a temporal transcription factor “matching code”, in which early-born ELs, Ladders, and Basins would all be derived from the same temporal transcription factor window. For NB3-5 and MNB, temporal transcription factor expression is only partially characterized. But, tantalizingly both early-born ELs and a subset of Ladders are born during a period in which their respective neuroblasts express the temporal transcription factor, Castor (Tsuji et al., 2008)(Kearney et al., 2004). A potential extrinsic mechanism is that early-born EL dendrites may provide some type of signal that promotes later born neurons to synapse. There is evidence for such communication among Drosophila nerve cord neurons (Valdes-Aleman et al., 2021). Finally, it will be interesting to understand if sequential assembly is an absolute requirement, or if instead it facilitates rapid, efficient, or robust circuit assembly.”

We love the idea of manipulating temporal transcription factors in the context of the early-born EL feed-forward circuit. Thank you for raising that suggestion. It is definitely on the short term “to do list”, but unfortunately it is outside the scope of this work.

Reviewer #3 (Recommendations for the authors):1) Figure 1: the ts-MARCM clones should be moved to the main results (from supplemental) since the birth order of neurons in the EL lineage is an important finding of the paper.

We moved the ts-MARCM clones from supplemental to main results. They now appear as Figure 2.

2) Lines 217-235, Figure 4: The network science-based approach is difficult to follow in the text and the figure. A clearer explanation of the methodology and results is needed to help the reader follow along. Figure 4B is particularly hard to understand.

Thank you for point out how brief our explanation was. We have updated the text. Lines 253-269 now read:

“To quantify patterns of connectivity we measured two features—similarity among left-right pairs of neurons and similarity among following pairs. An example of similarity among left-right pairs is A08j3 A1L and A08j3 A1R, both of which have significantly similar connectivity compared to A08x A1L (Figure 5A). In contrast, A08c A1L, but not A08c A1R, has significantly similar connectivity compared to A08o A1L (Figure 5A). In general, early-born ELs have significant similarities between left-right pairs of neurons (open purple circles in Figure 5B), whereas late-born ELs did not (open magenta circles, in Figure 5B). An example of similarity among following pairs is that A08c A1L has significantly similar connectivity compared to A08s A1L, which is adjacent to A08c in birth order (Figure 5A). Both early-born and late-born ELs have similarities among following pairs (filled circles in 5B). The idea that neurons within a temporal cohort are more similar to neurons next born in the sequence shows that within a temporal cohort there are graded transitions in connectivity patterns. Additionally, early-born ELs have similarities with many neurons within their temporal cohort, whereas late-born neurons only tend to be similar to neurons that were born next in the sequence (Figure 5A). We conclude temporal cohorts are not merely copies of each other and the computations performed by early-born and late-born ELs may differ.”

3) The calcium imaging data (Figure 5 J-L) could be moved to supplemental (especially since similar data was shown in a previous paper, though with a less sensitive tool).

We left the data in the main figure because we thought it illustrated the point we wanted to make.

4) Lineage-mapping and birth-dating the six neurons with significant synaptic input onto the EL-early neurons would be very helpful as this data would either validate or challenge the conclusions that (1) output neurons are born first and (2) only three lineages contribute to this circuit. As a first step, one could determine whether the cell bodies of the six unidentified neurons are clustered together (which would suggest that they are generated by the same neuroblast).

Thank you for this suggestion. We have included two new Figures, Figure 13 and 14 and a new Results section. Lines 508-574:

“Other neurons that are highly connected to early-born ELs come from multiple different lineages and most are born after early-born Els

In the *Drosophila* larval nerve cord, individual neurons make different numbers of synapses onto their various downstream partners. In this section, we focus on neurons that are highly connected to early-born ELs, but not in the three lineages previously discussed. We considered a neuron to be highly connected if it has 10 or more synapses onto early-born ELs (Table S6). We choose this criterion for highly connected neurons, first, because a majority of synapses onto early-born ELs are made by these neurons, and second, because these neurons are likely to be important drivers of the activity of early-born ELs. In the sections above, we focused on Ladder, Basin, and Early-born EL interneurons, which together with Chordotonal sensory neurons, account for the majority of neurons highly connected to early-born EL in A1 (Figure 13A, Table S6). From these data, we learned two “rules”. First, interneurons in the feed forward circuit come from temporal cohorts in three lineages. Second, output interneurons from one lineage are born before input interneurons from other lineages. Additionally, there are six other highly connected neurons (Figure 13A-C). Here, we characterize the developmental origins of these other neurons to understand the extent to which the rules apply.

First, we wanted to understand the extent to which the six other highly connected interneurons could be lineage-related. To assign lineage identity, we examined their cell body position and neurite fasciculation pattern. The six other neurons likely come from six different lineages (Figure 13B-C). Of these one is NB3-5, which generates Basin neurons. This means NB3-5 neurons contribute more synapses onto early-born ELs than we had previously appreciated. We conclude that together NB3-5, MNB, and NB3-3 generate a majority (75%) of interneurons that are highly connected to early-born ELs. However, the relationship is not exclusive, and interneurons from other lineages do also synapse multiple times onto early-born ELs.

Next, we wanted to understand the extent to which the six other highly connected interneurons could be born after early-born ELs. To do so, we examined the cell body position of each along the medio-lateral axis (Figure 13B’). Of the six, one had a medial cell body close to the neuropile, suggesting it was early-born within its lineage; four had cell body positions on or close to the lateral edge of the CNS, suggesting they were late-born with their respective lineages; and one had an intermediate cell body position that is difficult to interpret (Figure 13C). These data, along with our previous birth dating form Basins, Ladders, and ELs, suggest 70% of highly connected interneurons are born after early-born ELs. We conclude that large majority of highly connected neurons in this circuit are born after the circuit ouput neurons (early-born ELs), but that there are exceptions to this “rule”.

Finally, for one of the other highly connected other interneurons, we wanted to experimentally validate our lineage assignment and to more directly determine its birth time relative to early-born ELs. We chose to focus on neuron A03 upstream of A08m because it is inferred to be progeny of NB7-1, and NB7-1 is an extremely well-characterized lineage. NB7-1 delaminates early during neurogenesis (J. J. Broadus et al., 1995). The first six divisions of NB7-1 generate motor neurons and undifferentiated motor neuron siblings (Seroka and Doe, 2019)(Lacin et al., 2009). Then, NB7-1 divides approximately 14 more times, generating interneurons (Meng et al., 2019), one of which is thought to be neuron A083 upstream of A08m (Mark et al., 2021). We labeled neuron A03 upstream of A08m using a NB7-1-GAL4 driver line to create Multi Color Flip Out clones, which validates the lineage assignment (Figure 14A-B). Then, in the connectome, we measured the cortex neurite length of neuron A03 upstream of A08m and the other interneurons within the same hemilineage as a proxy for birth order (Figure 14C-D). At the earliest, neuron A03 upstream of A08m is the sixth-born interneuron, and because there are six divisions of NB7-1 before interneurons are made, neuron A03 upstream of A08m must be born on or after the 12th division (Figure 14E). To compare the birth timing of neuron A03 upstream of A08m, to the birth timing of early-born ELs, we present the following logic. NB3-3 and NB7-1 are generated at ~400 minutes and ~230 minutes, respectively (Doe, 1992). Neuroblasts divide every 45 minutes. And so, NB3-3 is four divisions behind of NB7-1 ([400 minutes -230 minutes]/45 divisions / minute = 3.8 divisions). Early-born ELs are made during the second to sixth divisions of NB3-3, which corresponds to the fifth to ninth divisions of NB7-1 (Figure 14E). Because NB7-1 produces neuron A03 upstream of A08m no earlier than the 12 division, we conclude it must be born after the early-born ELs. In addition, we noticed that three other interneurons in the same hemilineage as “A03 upstream of A08m” synapse onto early-born ELs, albeit less strongly. The cortex neurite length of these neurons shows they largely have birth times adjacent to neuron A03 upstream of A08m (Figure 14D), which is consistent with the idea that these neurons are a temporal cohort. Using the same logic as described earlier in this paragraph, we find all the NB7-1 interneurons that synapse onto early-born ELs are born after the early-born ELs themselves (Figure 14E). Thus, analysis of NB7-1 shows us that neurons from a third lineage follows the patterns identified for Ladder and Basin temporal cohorts. Furthermore, these data show additional temporal cohorts have neurons that synapse onto early-born ELs, but not as extensively as neurons in the Ladder, Basin, or early-born EL temporal cohorts.”

5) Are the temporal cohorts of the EL-early, Basin and Ladder neurons that wire together generated from the same tTF window? This would be very interesting to investigate as it could provide a model for how neurons find each other in the developing circuit.

This is a great question. Doing a proper investigation of this point is an important future direction. We addressed what is known in the Discussion (lines 769-775):

“A potential intrinsic mechanism is that of a temporal transcription factor “matching code”, in which early-born ELs, Ladders, and Basins would all be derived from the same temporal transcription factor window. For NB3-5 and MNB, temporal transcription factor expression is only partially characterized. But, tantalizingly both early-born ELs and a subset of Ladders are born during a period in which their respective neuroblasts express the temporal transcription factor, Castor (Tsuji et al., 2008)(Kearney et al., 2004).”

6) The definition of a feed-forward circuit should be included in the Results discussion when it is first reported (currently it is defined in the Discussion). This will help readers who may not be familiar with the term.

Thank you for yet another great point. We define the term in the introduction.

7) Transparent black boxes are visible in the panels of Fig6 C and D. This is likely a formatting error but should be corrected.

Figure 6 is now Figure 8. We wanted the black boxes to be visible so that it would be clear we were not manipulating data. We outlined them in white dashes to make them more obvious, and include an explicit statement in the Figure 8 figure legend.